# LEARNING COMPACT REGULAR DECISION PROCESSES USING PRIORS AND CASCADES

## ABSTRACT

In this work we study offline Reinforcement Learning (RL), and extend the previous work on learning Regular Decision Processes (RDPs), which are a class of non-Markovian environments, where the unknown dependency of future observations and rewards from the past interactions can be captured by some hidden finite-state automaton. We utilise the language metric introduced by Deb et al. (2025), and introduce a novel algorithm to learn a significantly more compact RDP with cycles, which are crucial for scaling to larger, more complex environments. Key to our results is a novel notion of *priors* for automaton learning, that allows us to exploit prior domain-related knowledge, used to factor out of the state space any feature that is known a priori. We validate our approach experimentally and provide a Probably Approximately Correct (PAC) analysis of our algorithm, showing it enjoys a sample complexity polynomial in the problem parameters.

## 1 INTRODUCTION

Reinforcement Learning (RL) is a family of algorithms for learning behaviour from repeated interactions with a stochastic dynamical system. A key assumption behind RL algorithms is the Markov property, which implies that the current observation and action are sufficient to predict the future evolution of the system (Puterman, 1994; Sutton et al., 1998). Though the Markov property is the basis for many efficient algorithms, there exist many applications –e.g., in robotics– where the Markov property does not hold. A common approach to deal with such cases is to consider a hidden state (Whitehead and Lin, 1995) to account for missing information. This is most notably studied in the context of Partially Observable Markov Decision Processes, or POMDPs (Kaelbling et al., 1998). Although the POMDP framework offers very expressive representations and is of great relevance in practice, it suffers from intractability in both planning and learning, and consequently the corresponding learning algorithms become impractical in large problems, unless some restrictive assumptions are imposed. An alternative recently proposed framework is Regular Decision Process, or RDP (Brafman and De Giacomo, 2019; 2024), wherein the past interaction history is compactly represented by a finite state automaton. In essence, an RDP is a special POMDP, whose hidden dynamics evolve according to some (unobservable) finite-state automaton featuring a controlled form of stochasticity, ensuring key favourable properties. Although RDPs by construction are less generic compared to POMDPs, they are computationally and statistically tractable. This has led to a growing interest in developing RDP learning algorithms from trajectories (Abadi and Brafman, 2020; Ronca and De Giacomo, 2021; Ronca et al., 2022; Cipollone et al., 2023; Deb et al., 2025).

We investigate offline RL in episodic RDPs, where the goal is to find a near-optimal policy from a dataset pre-collected using a behaviour policy. This problem was first studied by Cipollone et al. (2023), where a first algorithm with provable PAC-type performance guarantee in terms of sample complexity was proposed. Despite its appeal, this bound may imply a sample complexity growing exponentially in episode length in some problem instance. This was remedied in (Deb et al., 2025) through statistical tests defined via a novel metric called the *language metric*, specifically designed for traces, borrowing ideas from the theory of formal languages. However, the models presented in these papers consider an unstructured (i.e., atomic) hidden state modelling, which are incompetent to leverage some prior information one has about the structure of the hidden states.

In this paper, we extend previous work for RDP learning along two dimensions. The first contribution is to introduce *priors* for automaton learning. A prior is an automaton that incorporates prior

knowledge about a problem. Given one or more priors, an RDP can be expressed as the *cascade* composition of the prior automata and a domain-specific automaton. The notion of cascades were introduced in early works in Algebraic Automata Theory (Krohn and Rhodes, 1965; Ginzburg, 1968; Kaufman; Arbib, 1969), focussing primarily on decomposing semiautomata in terms of simpler or fundamental semiautomata, (prime semiautomata in the Krohn-Rhodes decomposition theorem Krohn and Rhodes (1965)). In our work, we explore the formulation of fundamental priors in terms of semiautomata, to be embedded into the target automata via the cascade product. This amounts to factoring out the features provided by the priors, and hence learning compact domain-specific automata. Notable priors include the *timestep prior* to factor out timesteps from the state space while still considering them, the *Markov prior* to specify that the previous observation may be relevant and avoid learning to remember it (notably this ensures that the domain-specific automaton will be a trivial single-state automaton if the RDP is in fact an MDP), and *spatial priors* that provide a description of the physical space of the domain and relieve the domain-specific automaton from having to learn it.

The second contribution is to allow cycles in the learned domain-specific automaton. In previous work the RDP states are organized in layers, one for each timestep. Introducing cycles can make the learned automaton significantly more compact, especially for episodic problems with long horizons. We identify conditions under which RDPs with cycles can be correctly learned, and demonstrate in experiments that the learned RDPs are often much smaller than in previous work. To learn RDPs with cycles our algorithm has to compare suffix distributions with different lengths, which is possible by exploiting the *language metric* (Deb et al., 2025). In addition to experiments, we perform a theoretical analysis of the sample complexity of our algorithm, showing it enjoys a sample complexity polynomial in the problem parameters.

## 1.1 RELATED WORK

Offline RL under the Markov property is by now well-established, and there exists a large and growing literature covering a broad range of MDP settings. In many settings, algorithms with optimal sample complexity bounds exist. To mention some notable studies, we refer to (Chen and Jiang, 2019; Jin et al., 2021; Li et al., 2024b; Rashidinejad et al., 2021; Uehara and Sun, 2022). Research on decision making under non-Markov assumption dates back to, at least, three decades ago; some early attempts include (Schmidhuber, 1990; Whitehead and Lin, 1995; Bacchus et al., 1996; Bakker, 2001). A classical and effective approach to tackle non-Markov problems was through considering hidden states (Whitehead and Lin, 1995), which related such problems to partially-observable problems. We discuss below the most relevant lines of research that can handle non-Markov problems, while excusing ourselves to give a through overview of all related developments.

**POMDPs, PSRs, and State Representation** There exist at least two major lines of research to handle hidden information states in the context of partial observability: POMDPs and state representations. Since RDPs are special POMDPs –with underlying dynamics evolving according to some finite state automaton–, RL algorithms for POMDPs also apply to RDPs. Unfortunately, tractable learning for general POMDPs remain to be an open problem, and to the best of our knowledge has only been achieved in subclasses such as ergodicity (Azizzadenesheli et al., 2016), undercomplete POMDPs (Guo et al., 2022; Jin et al., 2020), few-step decodability (Efroni et al., 2022; Krishnamurthy et al., 2016), or weakly-revealing (Liu et al., 2022). In this context, Hahn et al. (2024) introduces a generalization of RDPs with $\omega$-regular lookahead called Omega-Regular Decision Processes (ODPs) and provide classical complexity results. In the case of state representation, the most notable notion is Predictive State Representation (PSR) (Bowling et al., 2006; James and Singh, 2004; Kulesza et al., 2015; Singh et al., 2003), which provide general descriptions of dynamical systems; they capture POMDPs and therefore RDPs. However, existing work on PSRs (Zhan et al., 2023) rely on PSR-specific parameters and are therefore not directly applicable to RDPs.

**Reward Machines and RDPs** Some early work on non-Markov decision making restrict attention to non-Markov rewards, while assuming Markov dynamics. This is, for instance, considered in (Bacchus et al., 1996), where the reward function is specified in a temporal logic of the past. Revisiting this setting has led to some fast growing lines of research that notably include *reward machines* (Toro Icarte et al., 2018) and temporal logics of the future on finite traces (Brafman et al., 2018; Giacomo et al., 2020). A reward machine is a finite automaton (or transducer) used to specify a

non-Markovian reward function. Reward machines have been introduced in (Toro Icarte et al., 2018) along with an RL algorithm that assumes the reward machine to be known. There is a fast growing line of research on reward machines in a variety of settings; see, e.g., (Gaon and Brafman, 2020; Xu et al., 2020; Dohmen et al., 2022; Furelos-Blanco et al., 2023; Varricchione et al., 2024; Parać et al., 2024; Li et al., 2024a; Bourel et al., 2023). Reward machines have been generalised so as to predict observations as well (Toro Icarte et al., 2019; Hasanbeig et al., 2021), which makes them equivalent to RDPs—as mentioned above. Although some of these algorithms tackle the case of unobservable reward machines, they do not report performance guarantees on the proposed methods. Following their introduction by Brafman and De Giacomo (2019), RDPs were studied in the RL setting; in the online RL setting, some attempts include (Ronca and De Giacomo, 2021; Ronca et al., 2022; Abadi and Brafman, 2020). They are recently studied in the offline RL setting –the same setting considered here– following the work by Cipollone et al. (2023).

**Feature MDPs** Hutter (2009) introduces $\Phi$-MDPs, a generalization of POMDPs with feature maps. Concretely, the map $\Phi$ partitions the history space, and associates a state with each partition, and can be represented by trees, e.g. suffix trees (McCallum, 1996). As explained by Ron et al. (1993), a finite state automaton can be used to represent a prediction suffix tree, considering the nodes of the tree as the states of the automaton, where each state of the automaton is determined by the last $k$ inputs. In RDPs, the function $\bar{\tau}(h)$ is implicitly defined as a map from histories to RDP states, which is the RDP equivalent of the map $\Phi$ in feature MDPs.

## 2 PRELIMINARIES

**Notation** We use $\Delta(\mathcal{X})$ to denote the set of probability distributions over a set $\mathcal{X}$. A conditional probability distribution is a function $p : \mathcal{X} \to \Delta(\mathcal{Y})$ whose elements equal $p(y \mid x)$. We use $\mathbb{I}(E)$ to denote the indicator function of an event $E$. Given integers $m$ and $n$ such that $0 \le m \le n$, let $[\![m, n]\!] := \{m, \ldots, n\}$ and $[\![n]\!] := [\![1, n]\!]$. The notation $\tilde{\mathcal{O}}(\cdot)$ hides poly-logarithmic terms. All notations are collected and summarized in Appendix A.

### 2.1 LANGUAGE METRICS

The notion of language metric has been introduced by Deb et al. (2025), and here we present a close variant. Let $\Gamma$ be an alphabet, i.e. a finite set of symbols. Given a natural number $\ell \in \mathbb{N}$, let $\Gamma^\ell$ be the set of strings of symbols in $\Gamma$ of length $\ell$, and let $\Gamma^+ = \cup_{\ell=1}^\infty \Gamma^\ell$ be the set of non-empty strings of any length. The empty string is denoted $\varepsilon$. A *language* $X \subseteq \Gamma^+$ is a subset of non-empty strings. Let $\mathcal{X}$ be a set of languages. The *language metric* in $\mathcal{X}$ is a function $L_\mathcal{X} : \Delta(\Gamma^+) \times \Delta(\Gamma^+) \to \mathbb{R}$, on pairs of probability distributions $p, p' \in \Delta(\Gamma^+)$, defined as $L_\mathcal{X}(p, p') := \max_{X \in \mathcal{X}} |p(X) - p'(X)|$, where the probability of a language is $p(X) := \sum_{x \in X} p(x)$.

To learn cyclic automata in episodic RDPs we necessarily have to compare probability distributions over strings of different lengths. To do so we exploit the fact that the language metric $L_\mathcal{X}$ is a *pseudo-metric*: two different distributions $p \ne p'$ may satisfy $L_\mathcal{X}(p, p') = 0$. We are therefore interested in languages that are invariant to the string length. One such example is the family of languages that contain some pattern, e.g. any string that contains a given symbol $\gamma \in \Gamma$. Even if $p$ and $p'$ assign non-zero probability to strings of different lengths, we may still have $L_\mathcal{X}(p, p') = 0$.

### 2.2 EPISODIC DECISION PROCESSES AND REGULAR DECISION PROCESSES

An *episodic decision process* is a tuple $\mathbf{P} = \langle \mathcal{O}, \mathcal{A}, \mathcal{R}, \bar{T}, \bar{R}, H, \nu \rangle$, where $\mathcal{O}$ is a finite set of observations, $\mathcal{A}$ is a finite set of actions, $\mathcal{R} \subset [0, 1]$ is a finite set of rewards, $H > 0$ is an integer horizon, and $\nu \in \Delta(\mathcal{O})$ is an initial distribution on observations. We frequently consider the concatenation $\mathcal{AO}$ of the sets $\mathcal{A}$ and $\mathcal{O}$. Let $\mathcal{H}_t = (\mathcal{AO})^{t+1}$ be the set of histories of length $t + 1$, and let $h_{m:n} \in \mathcal{H}_{n-m}$ denote a history from time $m$ to time $n$, both included. Each action-observation pair $ao \in \mathcal{AO}$ in a history has an associated reward label $r \in \mathcal{R}$, which we write $ao/r \in \mathcal{AO}/\mathcal{R}$ with the understanding that the slash corresponds to string concatenation. A *trajectory* $e_{0:T}$ is the full history generated until (and including) time $T$.

We assume that a trajectory $e_{0:T}$ can be partitioned into *episodes* $e_{\ell:\ell+H} \in \mathcal{H}_H$ of length $H + 1$. In each episode $e_{0:H}$, $a_0$ is a dummy action and $o_0$ is sampled from the distribution $\nu$. The transition

function $\bar{T} : \mathcal{H} \times \mathcal{A} \to \Delta(\mathcal{O})$ and the reward function $\bar{R} : \mathcal{H} \times \mathcal{A} \to \Delta(\mathcal{R})$ depend on the current history in $\mathcal{H} = \cup_{t=0}^{H} \mathcal{H}_t$. Given $\mathbf{P}$, a *generic policy* is a function $\pi : (\mathcal{AO})^* \to \Delta(\mathcal{A})$ that maps trajectories to distributions over actions. The *value function* $V^\pi : [\![0, H]\!] \times \mathcal{H} \to \mathbb{R}$ of a policy $\pi$ is a mapping that assigns real values to histories. For $h \in \mathcal{H}$, it is defined as $V^\pi(H, h) := 0$ and $V^\pi(t, h) := \mathbb{E}\big[\sum_{i=t+1}^{H} r_i \,|\, h, \pi\big]$, for all timestep $t \in [\![0, H]\!]$, for all history $h \in \mathcal{H}_t$. For brevity, we write $V_t^\pi(h) := V^\pi(t, h)$. The *optimal value function* $V^*$ is defined as $V_t^*(h) := \sup_\pi V_t^\pi(h), \forall t \in [\![0, H]\!], \forall h \in \mathcal{H}_t$, where sup is taken over all policies $\pi : (\mathcal{AO})^* \to \Delta(\mathcal{A})$. Any policy achieving $V^*$ is called an *optimal policy*, which we denote by $\pi^*$; namely $V^{\pi^*} = V^*$. In what follows, we consider simpler policies of the form $\pi : \mathcal{H} \to \Delta(\mathcal{A})$ mapping finite histories to distributions over actions. Let $\Pi_\mathcal{H}$ denote the set of such policies. It can be shown that $\Pi_\mathcal{H}$ always contains an optimal policy, i.e. $V_t^*(h) := \max_{\pi \in \Pi_\mathcal{H}} V_t^\pi(h), \forall t \in [H], \forall h \in \mathcal{H}_t$. A policy $\widehat{\pi}$ is $\varepsilon$-optimal iff $\mathbb{E}_{h_0}[V_0^*(h_0) - V_0^{\widehat{\pi}}(h_0)] \leq \varepsilon$, where $h_0 = a_\perp o_0$ for some $o_0 \sim \nu$.

Each history $h \in \mathcal{H}_t$ and policy $\pi$ induces a probability distribution over suffixes $p_h^\pi \in \Delta(\Gamma^{H-t})$, where $\Gamma = \mathcal{AO}/\mathcal{R}$ is the alphabet of action-observation-reward triplets. Concretely, the probability of a suffix $e_{t+1:H} = a_{t+1} o_{t+1}/r_{t+1} \cdots a_H o_H/r_H$ is given by

$$p_h^\pi(e_{t+1:H}) = \prod_{i=t+1}^{H} \pi(a_i|h_{i-1}) \, \bar{T}(o_i|h_{i-1}, a_i) \, \bar{R}(r_i|h_{i-1}, a_i),$$

where $h_{i-1} = h a_{t+1} o_{t+1} \cdots a_{i-1} o_{i-1}$ for each $i \in [\![t+1, H]\!]$. Two histories $h$ and $h'$ are *equivalent* w.r.t. a class of policies $\Pi$ if $p_h^\pi = p_{h'}^\pi$ for every policy $\pi \in \Pi$; we write equivalence as $h \sim_\Pi h'$.
*Observation* 1. Specific policies may induce the same distribution for histories that are not equivalent. Namely, for a class of policies $\Pi$, and two histories $h$ and $h'$, a policy $\pi_1 \in \Pi$ may induce different distributions $p_h^{\pi_1} \neq p_{h'}^{\pi_1}$, while a second policy $\pi_2 \in \Pi$ may induce identical distributions $p_h^{\pi_2} = p_{h'}^{\pi_2}$ (as shown in Example 5, Appendix D).

**Episodic RDPs** We adopt the episodic variant of RDPs by Deb et al. (2025), a minor modification of the one by Cipollone et al. (2023). An *episodic regular decision process* is an episodic decision process $\mathbf{R} = \langle \mathcal{O}, \mathcal{A}, \mathcal{R}, \bar{T}, \bar{R}, H, \nu \rangle$ described by a *probabilistic-deterministic finite automaton*, or simply *automaton* for us, of the specific form $\mathbf{A} = \langle \mathcal{U}, \Sigma, \Omega, \tau, \theta, u_0 \rangle$ with $\mathcal{U}$ a finite set of states, $\Sigma = \mathcal{AO}$ a finite input alphabet composed of actions and observations, $\Omega$ a finite output alphabet, $\tau : \mathcal{U} \times \Sigma \to \mathcal{U}$ a transition function, $\theta : \mathcal{U} \to \Omega$ an output function, and $u_0 \in \mathcal{U}$ an initial state. Let $\tau^{-1}$ denote the inverse of $\tau$, i.e., $\tau^{-1}(u) \subseteq \mathcal{U} \times \mathcal{AO}$ is the subset of state-input pairs that map to $u \in \mathcal{U}$. An RDP $\mathbf{R}$ implicitly represents a function $\bar{\tau} : \mathcal{H} \to \mathcal{U}$ from histories in $\mathcal{H}$ to states in $\mathcal{U}$, recursively defined as $\bar{\tau}(h_0) := \tau(q_0, a_0 o_0)$ and $\bar{\tau}(h_t) := \tau(\bar{\tau}(h_{t-1}), a_t o_t)$. We use $A, O, R, U$ to denote the cardinality of $\mathcal{A}, \mathcal{O}, \mathcal{R}, \mathcal{U}$, respectively, and assume $H \geq 2$, $A \geq 2$ and $O \geq 2$.

The output function $\theta : \mathcal{U} \to \Omega$ maps the current state to an output in $\Omega$. The output space $\Omega = \Omega_\mathsf{o} \times \Omega_\mathsf{r}$ consists of a finite set of functions that specify the conditional probabilities of observations and rewards, of the form $\Omega_\mathsf{o} \subseteq \mathcal{A} \to \Delta(\mathcal{O})$ and $\Omega_\mathsf{r} \subseteq \mathcal{A} \to \Delta(\mathcal{R})$. For convenience, we often split the output function into two functions $\theta_\mathsf{o} : \mathcal{U} \times \mathcal{A} \to \Delta(\mathcal{O})$ and $\theta_\mathsf{r} : \mathcal{U} \times \mathcal{A} \to \Delta(\mathcal{R})$ specifying the conditional probabilities separately. The transition function and reward function of $\mathbf{R}$ are defined as $\bar{T}(o \mid h, a) = \theta_\mathsf{o}(o \mid \bar{\tau}(h), a)$ and $\bar{R}(r \mid h, a) = \theta_\mathsf{r}(r \mid \bar{\tau}(h), a)$ for each history $h \in \mathcal{H}$ and action-observation-reward triplet $ao/r \in \mathcal{AO}/\mathcal{R}$. An RDP is *minimal* if its automaton is minimal, i.e., without redundant states, and hence *unique*, cf. (Hartmanis and Stearns, 1966).

The class $\Pi_\mathbf{R}$ of policies acting according to the states of an RDP $\mathbf{R}$ is of particular importance. They are called *regular policies*, and they are defined as the policies $\pi : \mathcal{H} \to \Delta(\mathcal{A})$ satisfying the equality $\pi(h_1) = \pi(h_2)$ for all pairs of equivalent histories $h_1, h_2$ mapping to same state $u = \bar{\tau}(h) = \bar{\tau}(h')$. Hence, we can compactly define a regular policy as a function of the state, i.e., $\pi : \mathcal{U} \to \Delta(\mathcal{A})$. Regular policies exhibit *key properties*: (*P1*) under a regular policy, suffixes have the same probability of being generated for histories that map to the same state in $\mathcal{U}$; (*P2*) there exists at least one optimal policy that is regular; (*P3*) in the special case where an RDP is Markovian in both observations and rewards, it is sufficient for the states in $\mathcal{U}$ to track the observation in $\mathcal{O}$.

For RDPs, under regular policies, the notion of history equivalence admits an alternative form. Two histories $h$ and $h'$ are equivalent if and only if they map to the same state, i.e., $h \sim_{\Pi_\mathbf{R}} h' \Leftrightarrow \bar{\tau}(h) = \bar{\tau}(h') = u$ for $u \in \mathcal{U}$. In this setting, we can write $p_u^\pi$ in place of the identical distributions $p_h^\pi$ and $p_{h'}^\pi$. This shows that the meaning of a history is captured by the state $u = \bar{\tau}(h)$ the history maps to.

**Distinguishing RDP states** We will use language metrics $L_\mathcal{X}$ to learn RDPs from data. Thus we are interested in language sets $\mathcal{X}$ that correctly distinguish an RDP $\mathbf{R}$. For a given regular

policy $\pi$, a language set *distinguishes* an RDP $\mathbf{R}$ if: (i) For each pair of histories $h, h'$ such that $\bar{\tau}(h) = \bar{\tau}(h')$, $L_{\mathcal{X}}(p_h^\pi, p_{h'}^\pi) = 0$; (ii) for each pair of histories $h, h'$ such that $\bar{\tau}(h) \neq \bar{\tau}(h')$, $L_{\mathcal{X}}(p_h^\pi, p_{h'}^\pi) \geq \mu_{\mathcal{X}} > 0$. Two histories $h, h'$ s.t. $\bar{\tau}(h) = \bar{\tau}(h')$ may have different lengths. In this case we have $p_h^\pi \neq p_{h'}^\pi$, but $L_{\mathcal{X}}(p_h^\pi, p_{h'}^\pi) = 0$ may still hold due to $L_{\mathcal{X}}$ being a pseudo-metric. The quantity $\mu_{\mathcal{X}} \coloneqq \inf_{h, h' : \bar{\tau}(h) \neq \bar{\tau}(h')} L_{\mathcal{X}}(p_h^\pi, p_{h'}^\pi)$ is called the *distinguishability* of $\mathbf{R}$ under $\mathcal{X}$ and $\pi$.

**Automata cascades**  Representing automata with a state space $\mathcal{U}$ of atomic elements does not allow for specifying the complex meaning of a state and the single functionalities implemented by the transition function $\tau$ in order to perform state updates. Cascades offer a richer way to represent automata and overcome such limitations. A *cascade* is an automaton $\mathbf{C} = \langle \Sigma, \mathcal{U}, \tau, u_0, \Omega, \theta \rangle$ given by the composition $\mathbf{A}_1 \ltimes \cdots \ltimes \mathbf{A}_d$ where every $\mathbf{A}_i = \langle \Sigma_i, \mathcal{U}_i, \tau_i, u_0^i \rangle$ is a partial automaton that only specifies the components relevant to describe transitions (called a *semiautomaton*, following the terminology of automata theory). Every $\mathbf{A}_i$ is called a *component* of the cascade, and its input alphabet is $\Sigma_i \coloneqq \mathcal{U}_1 \cdots \mathcal{U}_{i-1} \Sigma$, allowing it to read the states of the preceding components in addition to inputs from $\Sigma$. Then, the states of $\mathbf{C}$ are given by the states of the single components, with $\mathcal{U} \coloneqq \mathcal{U}_1 \times \cdots \times \mathcal{U}_d$ and $u_0 \coloneqq \langle u_0^1, \ldots, u_0^d \rangle$, and the transition function is

$$\tau(u_1, \ldots, u_d, \sigma) \coloneqq \langle \tau_1(u_1, \sigma), \tau_2(u_2, u_1 \sigma), \ldots, \tau_d(u_d, u_1 \cdots u_{d-1} \sigma) \rangle,$$

where the transition function $\tau_i$ of the $i$-th cascade component is applied to the component's state $u_i$ and to the extended input $u_1 \cdots u_{i-1} \sigma$ containing the states of the preceding cascade components in addition to the input $\sigma$. We note that a component does not need to depend on all preceding components necessarily—in some cases, its transition function may ignore the state of some of the preceding components. This can be specified through the *cross-product* notation. For example, we can write $(\mathbf{A}_1 \times \mathbf{A}_2) \ltimes \mathbf{A}_3$ to say that $\mathbf{A}_2$ ignores the state of $\mathbf{A}_1$, and then $\mathbf{A}_3$ reads the state of both $\mathbf{A}_1$ and $\mathbf{A}_2$—note that parentheses are important to make it clear that we are not stating that $\mathbf{A}_3$ is independent of $\mathbf{A}_1$. We remark that cascades offer an advanced representation formalism–compared to conventional representations that are oblivious of the structure of states and transition function—as they allow for specifying how an automaton is realised by the composition of several components, each implementing a specific functionality, building on information already computed by the preceding components. This observation applies directly to the transition function $\tau$, and indirectly also to the output function $\theta$. In fact, the output function $\theta : \mathcal{U}_1 \times \cdots \times \mathcal{U}_d \to \Omega$ of a cascade is over a factored state space, which allows for richer descriptions that make it explicit how the function depends on the single state components. Note however that technical tools developed for learning Factored MDPs (e.g., Rosenberg and Mansour (2021); Strehl et al. (2007); Talebi et al. (2021); Tian et al. (2020)) do not carry over to our setting, because of unobservability of RDP states. Learning factored representations under partial observability is seldom studied in the literature, and the few existing work (e.g., Sallans (1999)) lack theoretical guarantees.

## 2.3   OFFLINE RL IN EPISODIC RDPS

Consider a batch dataset $\mathcal{D}$ comprising episodes sampled using an *admissible regular* behavior policy $\pi^{\mathsf{b}}$. Specifically, the $k$-th episode (or episode trace) in $\mathcal{D}$ is of the form $e_{0:H}^k = a_0^k o_0^k / r_0^k \cdots a_H^k o_H^k / r_H^k$ where, for each $t \in [\![H]\!]$,

$$o_0^k \sim \nu, \quad u_0^k = u_0, \quad a_t^k \sim \pi^{\mathsf{b}}(u_t^k), \quad o_t^k \sim \theta_{\mathsf{o}}(u_t^k, a_t^k), \quad r_t^k \sim \theta_{\mathsf{r}}(u_t^k, a_t^k), \quad u_{t+1}^k = \tau(u_t^k, a_t^k o_t^k).$$

The learner seeks an $\varepsilon$-optimal policy $\widehat{\pi}$ for a given accuracy $\varepsilon \in (0, H]$, using the smallest dataset $\mathcal{D}$ possible, without further exploration. More precisely, we aim at finding $\widehat{\pi}$ satisfying $V_0^*(h) - V_0^{\widehat{\pi}}(h) \leq \varepsilon$ for each $h \in \mathcal{H}$ with probability at least $1 - \delta$, using the smallest dataset $\mathcal{D}$ possible. We stress that in so doing $\pi^{\mathsf{b}}$ and underlying RDP states $u_t^k$ are unknown to the learner. It suffices to restrict attention to regular $\varepsilon$-optimal policies (cf. Proposition 5 in Deb et al. (2025)). However, some assumptions must be imposed on $\pi^{\mathsf{b}}$ to provably guarantee that an $\varepsilon$-optimal regular policy can be learned from $\mathcal{D}$.

Given a regular policy $\pi : \mathcal{U} \to \Delta(\mathcal{A})$, let $d_t^\pi \in \Delta(\mathcal{U} \times \mathcal{A}\mathcal{O})$ be the induced *occupancy*, i.e., a probability distribution over candidate states $u, ao \in \mathcal{U} \times \mathcal{A}\mathcal{O}$, recursively defined as

$$d_0^\pi(u_0, a_0 o_0) = \nu(o_0),$$

$$d_t^\pi(u_t, a_t o_t) = \textstyle\sum_{u, ao \in \tau^{-1}(u_t)} d_{t-1}^\pi(u, ao) \, \pi(a_t \mid u_t) \, \theta_{\mathsf{o}}(o_t \mid u_t, a_t), \quad \forall t \in [\![H]\!].$$

Of particular interest is the occupancy $d_t^* := d_t^{\pi^*}$ associated with an optimal policy $\pi^*$, which is unique if we assume that $\pi^*$ is unique. Likewise, let $d_t^{\mathsf{b}} := d_t^{\pi^{\mathsf{b}}}$ be the occupancy associated with $\pi^{\mathsf{b}}$. Since a state $u \in \mathcal{U}$ may appear at different time steps, we often abuse notation and write $d^{\mathsf{b}}(u, ao)$ or $d^*(u, ao)$ to denote the occupancy of $u, ao$ for the *first* timestep at which $u$ may appear.

As in offline RL in MDPs, it is necessary to control the mismatch in occupancy between the behavior policy $\pi^{\mathsf{b}}$ and the optimal policy $\pi^*$. Concretely, the single-policy RDP concentrability coefficient associated with RDP $\mathbf{R}$ and behavior policy $\pi^{\mathsf{b}}$ is defined as

$$C_{\mathbf{R}}^* = \max_{u, ao \in \mathcal{U} \times \mathcal{AO}} \frac{d^*(u, ao)}{d^{\mathsf{b}}(u, ao)} \,.$$

It is generally impossible to learn an RDP correctly from samples collected under a behaviour policy that does not have a finite concentrability coefficient, since this describes a situation where important states are not explored. Thus, we assume concentrability to be bounded away from infinity, $C_{\mathbf{R}}^* < \infty$, which further implies that for every $u, ao \in \mathcal{U} \times \mathcal{AO}$, $d^{\mathsf{b}}(u, ao) > 0$ whenever $d^*(u, ao) > 0$. In what follows $\mu_\mathcal{X}$ refers specifically to the distinguishability under the regular behavior policy $\pi^{\mathsf{b}}$.

## 3    Novel Techniques and Concepts

Equipped with the notions and definitions introduced in Section 2, we introduce two key notions that prove instrumental in the design of our proposed algorithm (Section 2.3). The first one deals with incorporating and leveraging some prior knowledge in RDPs, while the second characterises particularly-favourable cases for learning RDPs with priors, also extending the stationarity assumption in terms of timestep priors. We believe these notions could be of independent interest beyond RDPs.

### 3.1    Priors for RDPs

We introduce the novel notion of *priors* for RDPs, that allow for shaping the state space of an RDP with fundamental structures known a priori. This enables learning algorithms to focus on domain-specific aspects, relieving them from the burden of having to learn fundamental structures that are known to be present in a domain. A *prior* is an automaton without output components (a semiautomaton), $\mathbf{A}_{\mathsf{p}} = \langle \Sigma_{\mathsf{p}}, \mathcal{U}_{\mathsf{p}}, \tau_{\mathsf{p}}, u_0^{\mathsf{p}} \rangle$ with input alphabet $\Sigma_{\mathsf{p}} = \mathcal{AO}$, or alternatively $\Sigma_{\mathsf{p}} = \mathcal{U}_{\mathsf{p}}' \mathcal{AO}$ in the case it is part of a cascade where it depends on additional priors that precede it in the cascade and provide it with states from $\mathcal{U}_{\mathsf{p}}'$. Priors are included in the representation of an RDP by expressing its automaton $\mathbf{A}$ as a cascade $\mathbf{A} = \mathbf{A}_{\mathsf{p}} \ltimes \mathbf{A}_{\mathsf{r}}$ where $\mathbf{A}_{\mathsf{r}}$ is a second 'remainder' semiautomaton. In general, we can include several priors as $\mathbf{A} = \mathbf{A}_{\mathsf{p}}^1 \ltimes \cdots \ltimes \mathbf{A}_{\mathsf{p}}^m \ltimes \mathbf{A}_{\mathsf{r}}$. We can specify independence between some of the cascade components as, e.g., $\mathbf{A} = (\mathbf{A}_{\mathsf{p}}^1 \times \mathbf{A}_{\mathsf{p}}^2) \ltimes \mathbf{A}_{\mathsf{r}}$. Effectively, cascades allow for decomposing $\mathbf{A}$ into several components, each factoring out a specific feature implicit in the states of $\mathbf{A}$. The cascade decomposition focuses on states and transitions, but also provides a structured state space that allows for richer descriptions of the output function of $\mathbf{A}$. In fact, output functions will be over a factored state space $\mathcal{U}_{\mathsf{p}}^1 \times \cdots \times \mathcal{U}_{\mathsf{p}}^m \times \mathcal{U}_{\mathsf{r}}$ (abbreviated as $\mathcal{U}_{\mathsf{p}}^{1:m} \times \mathcal{U}_{\mathsf{r}}$), and they can be seen as functions of the overall state as in (a), or as functions of $\mathcal{U}_{\mathsf{r}}$ mapping to functions over the prior state space $\mathcal{U}_{\mathsf{p}}^{1:m}$ as in (b),

$$(\text{a}) \quad \theta : \left( \mathcal{U}_{\mathsf{p}}^{1:m} \times \mathcal{U}_{\mathsf{r}} \right) \to \left( \mathcal{A} \to \Delta(\mathcal{OR}) \right), \qquad (\text{b}) \quad \theta : \mathcal{U}_{\mathsf{r}} \to \left( \mathcal{U}_{\mathsf{p}}^{1:m} \to \left( \mathcal{A} \to \Delta(\mathcal{OR}) \right) \right).$$

Note that, although the output function of $\mathbf{A}$ has an extended domain, the automaton $\mathbf{A}$ still represents the functions $\bar{T}$ and $\bar{R}$ of the RDP over histories as usual. Specifically, the cascade decomposition only changes the way we express the (hidden) states of an RDP, that are now seen as consisting of several components focusing on specific aspects. It is also important to note that, although the factored state space may contain extra states compared to the standard state space consisting of atomic elements, this redundancy does not prevent the cascaded automaton from representing the RDP correctly, since redundant states can be 'collapsed' by the output function—formally, there may not be a bijection (isomorphism), but there is always an injection (homomorphism) that maps factored states to the corresponding atomic states.

Next we describe three of the most fundamental priors, and showcase their usage in RDPs.

**Markov priors** Markov priors allow for specifying that the previous observation may be a relevant feature in determining distributions over episode suffixes. Markov priors are simple semiautomata that store the previous observation. Specifically, the *Markov prior* for observations $\mathcal{O}$ is $\mathbf{M}_{\mathcal{O}} = \langle \mathcal{AO}, \mathcal{O} \cup \{\star\}, \tau_{\mathrm{o}}, \star \rangle$ where the initial state '$\star$' is an arbitrary element not in $\mathcal{O}$, and the transition function is the function $\tau_{\mathrm{o}}(o, ao') := o'$, that simply returns $o'$ disregarding $o$ and $a$. Including a Markov prior in the RDP automaton as $\mathbf{A} = \mathbf{M}_{\mathcal{O}} \ltimes \mathbf{A}_{\mathrm{r}}$ allows for factoring out the functionality of storing the previous observations, hence avoiding that this aspect is factored into the state space of $\mathbf{A}_{\mathrm{r}}$, which is left more compact and cleaner.

**Timestep priors** Timestep priors allow for specifying that the current timestep in an episode may be a relevant feature in determining distributions over episode suffixes. Timestep priors are simple semiautomata that count the number of timesteps elapsed. Specifically, the *timestep prior* for horizon $H$ is $\mathbf{T}_H = \langle \mathcal{AO}, [\![0, H]\!], \tau_{\mathrm{t}}, 0 \rangle$ where the transition function is defined as $\tau_{\mathrm{t}}(t, ao) := t + 1$. Including a timestep prior in the RDP automaton as $\mathbf{A} = \mathbf{T}_H \ltimes \mathbf{A}_{\mathrm{r}}$ allows for factoring out the functionality of keeping track of the current timestep, hence avoiding that this aspect is factored into the state space of $\mathbf{A}_{\mathrm{r}}$, which is left more compact and cleaner.

**Spatial priors** Spatial priors allow for describing the physical space (its geometry) of a domain, and specify that the current position in such space may be a relevant feature in determining distributions over episode suffixes. Automata allow for describing all finite spaces. A notable instance is the $m \times n$ *grid prior* for an RDP including motion actions $\mathcal{A}_{\mathrm{m}} = \{\rightarrow, \leftarrow, \uparrow, \downarrow\} \subseteq \mathcal{A}$, defined as $\mathbf{G}_{m \times n} = \langle \mathcal{AO}, [\![m]\!] \times [\![n]\!], \tau_{m \times n}, \langle x_0, y_0 \rangle \rangle$ with transition function $\tau_{m \times n}(x, y, ao)$ returning updated coordinates when $a$ is one of the motion actions.

**A notable case: RDPs with Markov and timestep priors** To convey a clearer idea of the effect of priors, we show explicitly what the automaton of an RDP looks like for the notable case when Markov and timestep priors are included at the same time. In particular, the two priors do not depend on each other, and hence they are composed as $\mathbf{T}_H \times \mathbf{M}_{\mathcal{O}}$. Then, the automaton of the RDP is expressed as $(\mathbf{T}_H \times \mathbf{M}_{\mathcal{O}}) \ltimes \mathbf{A}_{\mathrm{r}}$. The resulting state space is $\mathcal{U} = [\![0, H]\!] \times \mathcal{O} \times \mathcal{U}_{\mathrm{r}}$, and the transition function is

$$\tau(t, o, u_{\mathrm{r}}, ao') \,=\, \langle \tau_{\mathrm{t}}(t, ao'), \tau_{\mathrm{o}}(o, ao'), \tau_{\mathrm{r}}(u_{\mathrm{r}}, toao') \rangle \,=\, \langle t+1, o', u_{\mathrm{r}}' \rangle,$$

where $u_{\mathrm{r}}' = \tau_{\mathrm{r}}(u_{\mathrm{r}}, toao')$ is the result of applying the transition function $\tau_{\mathrm{r}}$ of $\mathbf{A}_{\mathrm{r}}$ to the previous state $u_{\mathrm{r}}$ and the extended input $toao'$, which includes the current timestep $t$ and the previous observation $o$, in addition to the current action $a$ and observation $o'$.

## 3.2 Partial independence from priors and semi-stationarity

In some special cases, the domain-specific automaton can be learned without considering priors explicitly at learning time. Let us consider an RDP expressed as a cascade $\mathbf{A}_{\mathsf{p}} \ltimes \mathbf{A}_{\mathrm{r}}$ where $\mathbf{A}_{\mathsf{p}}$ is a prior and $\mathbf{A}_{\mathrm{r}}$ is a domain-specific automaton. This yields a state space $\mathcal{U} = \mathcal{U}_{\mathsf{p}} \times \mathcal{U}_{\mathrm{r}}$, and hence an output function of the form $\theta : \mathcal{U}_{\mathsf{p}} \times \mathcal{U}_{\mathrm{r}} \to (\mathcal{A} \to \Delta(\mathcal{OR}))$. Intuitively, this cascade representation amounts to a factoring out the cascade features from $\mathbf{A}$. Then, the special case when priors can be considered separately is captured by the following notion.

**Definition 1.** An RDP represented by the cascade $\mathbf{A}_{\mathsf{p}} \ltimes \mathbf{A}_{\mathrm{r}}$ is *partially independent from priors* when the following conditions hold: (I) the two cascade components are independent, $\mathbf{A} = \mathbf{A}_{\mathsf{p}} \times \mathbf{A}_{\mathrm{r}}$, (II) the observation function $\theta_{\mathsf{o}}$ of $\mathbf{A}$ can be expressed as the product of two independent functions as $\theta_{\mathsf{o}}(o \,|\, u_{\mathsf{p}}, u_{\mathrm{r}}, a) = \theta_{\mathsf{o}}^{\mathsf{p}}(o \,|\, u_{\mathsf{p}}, a) \cdot \theta_{\mathsf{o}}^{\mathrm{r}}(o \,|\, u_{\mathrm{r}}, a)$, and (III) the reward function $\theta_{\mathrm{r}}$ can be expressed as the product of two independent functions as $\theta_{\mathrm{r}}(r \,|\, u_{\mathsf{p}}, u_{\mathrm{r}}, a) = \theta_{\mathrm{r}}^{\mathsf{p}}(r \,|\, u_{\mathsf{p}}, a) \cdot \theta_{\mathrm{r}}^{\mathrm{r}}(r \,|\, u_{\mathrm{r}}, a)$. When Conditions (I) and (III) hold, we say the RDP is *partially independent from priors w.r.t. rewards*. When an RDP is partial independent from a timestep prior $\mathbf{T}_H$, we say the RDP is *semi-stationary*.

The definition applies to the case of multiple priors, as they can all be seen as part of $\mathbf{A}_{\mathsf{p}}$. Partial independence is important as it enables learning the domain-specific automaton $\mathbf{A}_{\mathrm{r}}$ while ignoring learning the prior $\mathbf{A}_{\mathsf{p}}$, since states $u_{\mathrm{r}} = \bar{\tau}_{\mathrm{r}}(h)$ and their transition function $\tau_{\mathrm{r}}$ can be learned by checking similarity of the distributions they induce on episode suffixes, which are independent of any feature provided by the priors. If independence is only w.r.t. rewards, only the reward function can be captured correctly by a cascade where independence from priors is included, which can still be useful to learn optimal policies.

Next we showcase the above notions through an example.

**Example 1.** The T-maze of length $N$ and horizon $H$ (Deb et al., 2025), when represented as $\mathbf{T}_H \ltimes \mathbf{A}_r$ is partially independent from the timestep prior $\mathbf{T}_H$, or semi-stationary. Furthermore, when represented as $(\mathbf{T}_H \times \mathbf{G}_{3\times(N+1)}) \ltimes \mathbf{A}_r$, with $\mathbf{G}_{3\times(N+1)}$ the grid prior, the RDP is partially independent from both priors w.r.t. rewards only. Further details are deferred to Appendix D.3.

## 4 ALGORITHM AND PAC ANALYSIS

In this section we present ADACT–L, our algorithm for learning RDPs with priors and cycles. The algorithm assumes that we are provided with a prior automaton $\mathbf{A}_p = \langle \Sigma, \mathcal{U}_p, \tau_p, u_p^0 \rangle$, and the aim is to learn a problem-specific automaton $\mathbf{A}_r = \langle \Sigma, \mathcal{U}_r, \tau_r, u_r^0 \rangle$ such that the complete RDP is expressed as a cascade $\mathbf{A} = \mathbf{A}_p \ltimes \mathbf{A}_r$. For this purpose, the transition function $\tau_r : \mathcal{U}_r \times \mathcal{U}_p\,\Sigma \to \mathcal{U}_r$ incorporates the states of the prior automaton as part of its input. We remark that the prior automaton $\mathbf{A}_p$ could itself be a cascade of automata, and that the algorithm can learn an RDP without prior knowledge by defining a prior automaton $\mathbf{A}_p$ with a single state.

Intuitively, ADACT–L learns an RDP $\mathbf{A}$ with composite states $u_p u_r$ in a breadth-first manner starting from $u_p^0 u_r^0$. To represent a transition from $u_p u_r$ to $u_p' u_r'$ as a result of observing $ao$, it is sufficient to define $\tau_r(u_r, u_p ao) = u_r'$, since the transition $\tau_p(u_p, ao) = u_p'$ is handled separately by the prior $\mathbf{A}_p$. The algorithm is based on the fact that the transition function $\tau_r$ is invariant to the identity of $u_r$ in composite states $u_p u_r$ as long as the initial state $u_p^0$ of $\mathbf{A}_p$ is always paired with the initial state $u_r^0$ of $\mathbf{A}_r$. For example, let $u_r^1$ and $u_r^2$ be two states of $\mathbf{A}_r$ and let $u_p$ be a state of $\mathbf{A}_p$. We can construct an equivalent automaton by swapping the definitions of $\tau_r(u_r^1, u_p \cdot)$ and $\tau_r(u_r^2, u_p \cdot)$ and changing the definition of $\tau_r(\cdot, u_p' ao)$ from $u_r^1$ to $u_r^2$ or vice versa whenever $\tau_p(u_p', ao) = u_p$.

As a consequence of the above fact, when we discover a new composite state $u_p' u_r'$, the identity of $u_r'$ can be arbitrary. In the algorithm, we simply choose $u_r'$ as next available state in $\mathcal{U}_r$ for $u_p'$. To do so, we assume that we have access to a sequence of states $u_r^0, u_r^1, u_r^2, \ldots$ and for each prior state $u_p' \in \mathcal{U}_p$ we remember the index $i(u_p') \geq 0$ of the next available state in $\mathcal{U}_r$. This also allows us to iterate over all existing composite states involving $u_p'$ (line 10). We also remember the first timestep $t(u_p u_r)$ of each state pair in order to add all suffixes of the same length to the associated multiset (line 17).

---

**Function** ADACT–L($\mathcal{U}_r$, $\mathcal{D}$, $\mathbf{A}_p$, $\delta$)

**Input:** Automaton states $\mathcal{U}_r = \{u_r^0, u_r^1, u_r^2, \ldots\}$, dataset $\mathcal{D}$, prior automaton $\mathbf{A}_p = \langle \Sigma, \mathcal{U}_p, \tau_p, u_p^0 \rangle$, $\delta$
**Output:** Transition function $\tau_r : \mathcal{U}_r \times \mathcal{U}_p\, \mathcal{A}\mathcal{O} \to \mathcal{U}_r$

1 **foreach** $u_p \in \mathcal{U}_p$ **do** $i(u_p) \leftarrow 0$
2 $Q \leftarrow \{u_p^0 u_r^0\}$                  // queue data structure containing $u_p^0 u_r^0$
3 $i(u_p^0) \leftarrow 1, t(u_p^0 u_r^0) \leftarrow 0, \mathcal{Z}(u_p^0 u_r^0) \leftarrow \mathcal{D}$
4 **while** $Q$ *is not empty* **do**
5      dequeue $u_p u_r$ from $Q$               // next joint state
6      **for** $ao \in \mathcal{A}\mathcal{O}$ **do**
7          $u_p' \leftarrow \tau_p(u_p, ao)$            // next prior state
8          $\mathcal{Z}(ao) \leftarrow \{e_{t+1:H} \mid ao/re_{t+1:H} \in \mathcal{Z}(u_p u_r)\}$     // compute suffixes
9          $j \leftarrow i(u_p')$
10          **for** $k = 0, \ldots, i(u_p') - 1$ **do**
11             **if** *not* TESTDISTINCT($\mathcal{Z}(u_p' u_r^k), \mathcal{Z}(ao), \delta$) **then** $j \leftarrow k$
12          **end**
13          $\tau_r(u_r, u_p ao) \leftarrow u_r^j$         // define transition function
14          **if** $j = i(u_p')$ **then**
15             enqueue $u_p' u_r^j$ in $Q$
16             $i(u_p') \leftarrow j + 1, t(u_p' u_r^j) \leftarrow t(u_p u_r) + 1, \mathcal{Z}(u_p' u_r^j) \leftarrow \mathcal{Z}(ao)$
17          **else if** $t(u_p' u_r^j) = t(u_p u_r) + 1$ **then** $\mathcal{Z}(u_p' u_r^j) \leftarrow \mathcal{Z}(u_p' u_r^j) \cup \mathcal{Z}(ao)$
18      **end**
19 **end**
20 **return** $\tau_r$
21 **Function** TESTDISTINCT($\mathcal{Z}_1$, $\mathcal{Z}_2$, $\delta$)
22      **return** $L_\mathcal{X}(\mathcal{Z}_1, \mathcal{Z}_2) \geq \sqrt{\log(2|\mathcal{X}|/\delta)/\min(|\mathcal{Z}_1|, |\mathcal{Z}_2|)}$     // statistical test

---

In Appendix B we prove the following sample complexity bound for ADACT–L.

**Theorem 1.** ADACT–L$(\mathcal{D}, \delta)$ *returns a minimal automaton* $\mathbf{A_r}$ *with probability at least* $1 - 2AOUU_p\, \delta$ *when using a language set* $\mathcal{X}$ *that distinguishes* $\mathbf{A_p} \bowtie \mathbf{A_r}$ *under the behavior policy* $\pi^{\mathbf{b}}$ *with associated distinguishability* $\mu_{\mathcal{X}}$ *and the size of the dataset* $\mathcal{D}$ *is at least*

$$|\mathcal{D}| \geq \widetilde{\mathcal{O}} \left( \frac{C^*_{\mathbf{R}} \log(1/\delta) \log |\mathcal{X}|}{d^*_m \cdot \mu^2_{\mathcal{X}}} \right), \qquad \text{with} \quad d^*_m := \min_{u,ao} d^*(u, ao).$$

In Appendix C we prove that a version of ADACT–L which returns an approximately optimal policy achieves an improved sample complexity. Further, we refer to Figure 1 in Appendix E, which illustrates the various steps of the RDP learning pipeline.

## 5 EXPERIMENTAL EVALUATION

We conduct numerical experiments to further demonstrate the performance and properties of ADACT–L. We present our results for five familiar domains in the literature of POMDPs and RDPs: Corridor (Ronca and De Giacomo, 2021), T-maze(c) (Bakker, 2001), Cookie (Toro Icarte et al., 2019), Cheese (McCallum, 1992) and Mini-hall (Littman et al., 1995), and summarize our results in Table 1. We compare against FlexFringe (Baumgartner and Verwer, 2023), a state-of-the-art algorithm for learning probabilistic-deterministic finite automata, which includes RDPs as a special case, and ADACT-H (Deb et al., 2025). FlexFringe can learn RDPs with cycles, but includes several heuristics that do not preserve high-probability sample complexity guarantees. ADACT-H learns RDPs without cycles. The proposed algorithm ADACT–L can learn cycles in addition to providing sample complexity guarantees. In all experiments we use a Markov prior and a language set $\mathcal{X}$ consisting of one language per action-observation-reward triplet, containing all strings of any length that includes the triplet. This language set may only learn an approximate RDP in some domains.

From our results in Table 1, we can see that ADACT–L learns much smaller automata, while also achieving the highest average reward. In T-maze(c), FlexFringe fails to find the optimal policy, since the heuristics defined for FlexFringe are not optimized to preserve reward. In the domains Cheese and Minihall, all the algorithms fail to learn the optimal policy owing to the complexity of the POMDP environments; however, ADACT–L outperforms the other approaches by getting a higher average reward as well as learning significantly smaller automata.

| Name | $H$ | FlexFringe | | | ADACT-H | | | ADACT–L | | |
|---|---|---|---|---|---|---|---|---|---|---|
| | | $U$ | $r$ | time | $U$ | $r$ | time | $U$ | $r$ | time |
| Corridor | 5 | 11 | **1.0** | 0.03 | 11 | **1.0** | **0.01** | **3** | **1.0** | **0.01** |
| T-maze(c) | 5 | 29 | 0.0 | 0.11 | 18 | **1.0** | 0.26 | **5** | **1.0** | **0.15** |
| Cookie | 9 | 220 | **1.0** | 0.36 | 91 | **1.0** | **0.08** | **11** | **1.0** | **0.08** |
| Cheese | 6 | 669 | $0.69 \pm .04$ | 19.28 | 1178 | $0.87 \pm .03$ | 12.11 | **85** | **0.89 ± .04** | **7.27** |
| Mini-hall | 15 | 897 | $0.33 \pm .04$ | 25.79 | 6098 | $0.86 \pm .03$ | 29.90 | **65** | **0.87 ± .04** | **25.18** |

Table 1: For each domain, $H$, $U$ are the horizon and the number of states in the learned automaton respectively, $r$ is the average normalised reward (over 100 episodes) of the derived policy, and 'time' is the running time in seconds of automaton learning. Best results emphasised in bold.

## 6 CONCLUSIONS

In this work, we introduce a novel algorithm ADACT–L utilizing the language metric introduced by Deb et al. (2025), which allows us to learn a significantly smaller RDP with cycles, and also identify conditions under which RDPs with cycles can be correctly learned which makes it possible to scale to larger and more complex domains. Further to exploit domain-related knowledge, we also introduce the notion of *priors* for automaton learning, that can be used to factor out of the state space any feature that is known a priori. We further validate our approach experimentally over five familiar domains in the POMDP and RDP literature, and compare the performance of our algorithm to FlexFringe, a state-of-the-art algorithm for learning PDFA. Finally, as future work, we plan to explore the approximate version of our algorithm and also to extend our work to the online setting.

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

# A  NOTATION

| | |
|---|---|
| $\Delta(\mathcal{X})$ | The probability distribution over set $\mathcal{X}$ |
| $p : \mathcal{X} \to \Delta(\mathcal{Y})$ | A conditional probability distribution whose elements are $p(y \mid x)$ |
| $\mathbb{I}(x)$ | The indicator function of event $x$ |
| $\Gamma$ | An alphabet or a finite set of symbols |
| $\Gamma^\ell$ | A set of strings of symbols in $\Gamma$ of length $\ell$, where $\ell \in \mathbb{N}$ |
| $\Gamma^+$ | A set of non-empty strings of symbols in $\Gamma$ of any length, given by $\Gamma^+ = \cup_{\ell=1}^\infty \Gamma^\ell$ |
| $X$ | A language, which is a subset of non-empty strings, given by $X \subseteq \Gamma^+$ |
| $\mathcal{X}$ | A set of languages |
| $L_{\mathcal{X}} : \Delta(\Gamma^+) \times \Delta(\Gamma^+) \to \mathbb{R}$ | A function that maps pairs of probability distributions $p, p' \in \Delta(\Gamma^+)$ to a real number. |
| **Episodic Decision Processes** | |
| $\mathbf{P}$ | An episodic decision process given by the tuple $\langle \mathcal{O}, \mathcal{A}, \mathcal{R}, \bar{T}, \bar{R}, H, \nu \rangle$ |
| $\mathcal{O}$ | A finite set of observations |
| $\mathcal{A}$ | A finite set of actions |
| $\mathcal{R} \subset [0,1]$ | A finite set of rewards |
| $H$ | Integer horizon where $H > 0$ |
| $\nu \in \Delta(\mathcal{O})$ | Initial distribution over observations |
| $\mathcal{H}_t$ | Set of histories of length $t+1$ given by $\mathcal{H}_t = (\mathcal{A}\mathcal{O})^{t+1}$ where $\mathcal{A}\mathcal{O}$ is the concatenation of the sets $\mathcal{A}$ and $\mathcal{O}$ |
| $\mathcal{H}$ | Set of histories of lengths up to $H+1$, given by $\mathcal{H} = \cup_{t=0}^H \mathcal{H}_t$ |
| $h_{m:n} \in \mathcal{H}_{n-m}$ | A history from time $m$ to time $n$ |
| $ao/r$ | String concatenation of $a \in \mathcal{A}$, $o \in \mathcal{O}$, and reward label $r \in \mathcal{R}$ |
| $e_{m:n}$ | An episode of length $n - m + 1$, such that $e_{m:n} \in (\mathcal{A}\mathcal{O}\mathcal{R})^{n-m+1}$, and the trajectory $e_{0:T}$ is the full episode generated until (and including) time $T$. |
| $\bar{T} : \mathcal{H} \times \mathcal{A} \to \Delta(\mathcal{O})$ | Transition function maps current history and action to a distribution over observations |
| $\bar{R} : \mathcal{H} \times \mathcal{A} \to \Delta(\mathcal{R})$ | Reward function maps current history and action to a distribution over rewards |
| $\pi : (\mathcal{A}\mathcal{O})^* \to \Delta(\mathcal{A})$ | A generic policy tha maps trajectories to distributions over actions |
| $V_t^\pi(h)$ | The value function of policy $\pi$ that assigns real values to histories, where $t \in [\![0, H]\!]$ and $h \in \mathcal{H}$ |

## Regular Decision Processes

| | |
|---|---|
| $\mathbf{R}$ | An episodic regular decision process, given by the tuple $\langle \mathcal{O}, \mathcal{A}, \mathcal{R}, \bar{T}, \bar{R}, H, \nu \rangle$ |
| $\mathbf{A}$ | A probabilistic-deterministic finite automaton, referred to as *automaton*, given by the tuple $\mathbf{A} = \langle \mathcal{U}, \Sigma, \Omega, \tau, \theta, u_0 \rangle$ |
| $\mathcal{U}$ | A finite set of states of an automaton |
| $\Sigma$ | A finite input alphabet composed of actions and observations, given as $\Sigma = \mathcal{AO}$ |
| $\Omega$ | A finite output alphabet |
| $\tau : \mathcal{U} \times \Sigma \to \mathcal{U}$ | A transition function that maps a state and an input to a state $u \in \mathcal{U}$ |
| $\theta : \mathcal{U} \to \Omega$ | An output function of the automaton |
| $\theta_{\mathsf{o}} : \mathcal{U} \times \mathcal{A} \to \Delta(\mathcal{O})$ | Output function maps state and action to distribution over observations |
| $\theta_{\mathsf{r}} : \mathcal{U} \times \mathcal{A} \to \Delta(\mathcal{R})$ | Output function maps state and action to distribution over rewards |
| $u_0 \in \mathcal{U}$ | Initial state of the automaton |
| $\tau^{-1}(u) \subseteq \mathcal{U} \times \mathcal{AO}$ | Inverse of the transition function $\tau$ |
| $\Pi_{\mathbf{R}}$ | Class of regular policies which can be defined as $\pi : \mathcal{U} \to \Delta(\mathcal{A})$ |
| $\mathbf{C}$ | An automaton given by the composition of $d$ partial automatons $\mathbf{A}_1 \ltimes \cdots \ltimes \mathbf{A}_d$ |
| $\pi^{\mathsf{b}}$ | An admissible regular behavior policy used to collect the dataset |
| $\mathcal{D}$ | A dataset collected by the policy $\pi^{\mathsf{b}}$ |
| $d_t^\pi \in \Delta(\mathcal{U} \times \mathcal{AO})$ | The induced occupancy of policy $\pi$ over $u, ao \in \mathcal{U} \times \mathcal{AO}$ |
| $C_{\mathbf{R}}^*$ | The single-policy RDP concentrability coefficient associated with RDP $\mathbf{R}$ |
| $\mathbf{M}_{\mathcal{O}}$ | A Markov prior given by the tuple $\langle \mathcal{AO}, \mathcal{O} \cup \{\star\}, \tau_{\mathsf{o}}, \star \rangle$ where $\star$ is an arbitary intial state not in $\mathcal{O}$ |
| $\mathbf{T}_H$ | A timestep prior given by $\langle \mathcal{AO}, [\![0, H]\!], \tau_{\mathsf{t}}, 0 \rangle$ |
| $\tau_{\mathsf{t}} : T \times \mathcal{AO} \to T$ | The transition function for a time step prior given as $\tau_{\mathsf{t}}(t, ao) \coloneqq t + 1$ |
| $\mathbf{G}_{m \times n}$ | A spatial prior given by $\langle \mathcal{AO}, [\![m]\!] \times [\![n]\!], \tau_{m \times n}, \langle x_0, y_0 \rangle \rangle$, defined for a $m \times n$ grid |
| $\tau_{m \times n} : [\![m]\!] \times [\![n]\!] \times \mathcal{AO} \to [\![m]\!] \times [\![n]\!])$ | The transition function for a spatial prior defined for a $m \times n$ grid |

# B   Technical Lemmas

The technical lemmas are reformulated from (Deb et al., 2025) for our setting. Following the proof-structure, we first provide the high probability upper bound on the language metric $L_{\mathcal{X}}$ adapted to our setting.

**Lemma 2.** *Let $\mathcal{X}$ be a language set. Given a candidate state $u, ao \in \mathcal{U} \times \mathcal{AO}$ and a multiset $\mathcal{Z}(uao)$ of suffixes in $\Gamma^+$, with probability at least $1 - \delta$ the language metric $L_{\mathcal{X}}$ satisfies*

$$L_{\mathcal{X}}(\widehat{p}_{uao}, p_{uao}) \leq \sqrt{\frac{\log(2|\mathcal{X}|/\delta)}{2|\mathcal{Z}(uao)|}},$$

*where $p_{uao} \in \Delta(\Gamma^+)$ is the true distribution on suffixes induced by the candidate $uao$ and the behavior policy $\pi^{\mathsf{b}}$, and $\widehat{p}_{uao} \in \Delta(\Gamma^+)$ is the empirical estimate on suffixes induced by $\mathcal{Z}(uao)$.*

*Proof.* Let $p_{uao}(X) = \sum_{x \in X} p_{uao}(x)$ be the true probability of each language language $X \in \mathcal{X}$, and let $\widehat{p}_{uao}(X) = \sum_{x \in \mathcal{Z}(uao)} \mathbb{I}(x \in X)/|\mathcal{Z}(uao)|$ be the empirical estimate of $p_{uao}(X)$. Following Hoeffding's inequality we get

$$\mathbb{P}\left(|\widehat{p}_{uao}(X) - p_{uao}(X)| > \sqrt{\frac{\log(2/\delta_{\mathsf{s}})}{2|\mathcal{Z}(uao)|}}\right) \leq \delta_{\mathsf{s}}.$$

Choosing $\delta_{\mathsf{s}} = \delta/|\mathcal{X}|$ and taking a union bound implies that $L_{\mathcal{X}}$ satisfies

$$L_{\mathcal{X}}(\widehat{p}_{uao}, p_{uao}) = \max_{X \in \mathcal{X}} |\widehat{p}_{uao}(X) - p_{uao}(X)| \leq \sqrt{\frac{\log(2|\mathcal{X}|/\delta)}{2|\mathcal{Z}(uao)|}}$$

with probability $1 - |\mathcal{X}|\delta_{\mathsf{s}} = 1 - \delta$, which completes the proof. $\qquad\qquad\square$

Next, we define an associated event $\mathcal{E}_{\mathcal{X}}$ to correctly bound the language metric $L_{\mathcal{X}}$ for all candidate states:

$$\mathcal{E}_{\mathcal{X}} = \left\{ \forall u, ao \in \mathcal{U} \times \mathcal{AO} : L_{\mathcal{X}}(\widehat{p}_{uao}, p_{uao}) \leq \sqrt{\frac{\log(2|\mathcal{X}|/\delta)}{2|\mathcal{Z}(uao)|}} \right\}.$$

We next prove a high-probability sample complexity bound for accurately estimating the occupancy $d^{\mathsf{b}}(u, ao)$ of each candidate state. Let $\widehat{d}(uao)$ be the empirical occupancy of $uao$. Given a number of episodes $N$, an empirical Bernstein inequality yields

$$\mathbb{P}\left(\left|\widehat{d}(uao) - d^{\mathsf{b}}(u, ao)\right| > \sqrt{\frac{2\widehat{d}(uao)\log(4/\delta)}{N}} + \frac{14\log(4/\delta)}{3N}\right) \leq \delta. \tag{1}$$

We can next define $G_{\delta}$ as the function for the bound in the empirical Bernstein inequality where $\delta$ is the given failure probability, given by

$$G_{\delta}(\widehat{d}, N) = \sqrt{\frac{2\widehat{d}\log(4/\delta)}{N}} + \frac{14\log(4/\delta)}{3N}$$

where $G_{\delta}$ is monotonically increasing in $\widehat{d}$ and monotonically decreasing in $N$. We can further define an associated event $\mathcal{E}_B$ to correctly bound $|\widehat{d}(uao) - d^{\mathsf{b}}(u, ao)|$ for all $hao$:

$$\mathcal{E}_B = \left\{ \forall u, ao \in \mathcal{U} \times \mathcal{AO} : \left|\widehat{d}(uao) - d^{\mathsf{b}}(u, ao)\right| \leq G_{\delta}(\widehat{d}(uao), N) \right\}.$$

The following lemma shows that we can control the number of episodes $N$ to obtain an upper bound on the function $G_{\delta}$.

**Lemma 3.** *For fixed probabilities $\delta$ and $\widehat{d}$, if $N \geq 16\log(4/\delta)/\widehat{d}$ it holds that $3G_{\delta}(\widehat{d}, N) < 2\widehat{d}$.*

*Proof.* We first show that the inequality holds for $N = 16 \log(4/\delta)/\widehat{d}$. In this case we have

$$3G_\delta(\widehat{d}, N) = 3\sqrt{\frac{2\widehat{d}^2 \log(4/\delta)}{16 \log(4/\delta)} + \frac{14\widehat{d} \log(4/\delta)}{16 \log(4/\delta)}} = \left(\frac{3}{\sqrt{8}} + \frac{14}{16}\right)\widehat{d} < 2\widehat{d}.$$

The case $N > 16 \log(4/\delta)/\widehat{d}$ follows from the fact that $G_\delta$ is monotonically decreasing in $N$. $\quad\square$

Since $\widehat{d}(uao) = |\mathcal{Z}(uao)|/N$ implies $N = |\mathcal{Z}(uao)|/\widehat{d}(uao)$, we obtain the following corollary.

**Corollary 4.** *Under event $\mathcal{E}_B$, if $|\mathcal{Z}(uao)| \geq 16 \log(4/\delta)$, it holds that $|\widehat{d}(uao) - d^{\mathsf{b}}(u, ao)| \leq 2\widehat{d}(uao)/3$.*

We show that under event $\mathcal{E}_B$, we can choose the sample complexity $N$ to ensure that we obtain at least a certain number of elements in $\mathcal{Z}(uao)$.

**Lemma 5.** *Given a candidate state $u, ao \in \mathcal{U} \times \mathcal{AO}$, under event $\mathcal{E}_B$, it holds that $|\mathcal{Z}(uao)| \geq b \log(4/\delta)$ if the sample complexity $N$ satisfies*

$$N \geq \frac{\log(4/\delta)}{d^{\mathsf{b}}(u, ao)} \left(2b + 31/6\right).$$

*Proof.* Letting $M = |\mathcal{Z}(uao)|$, due to event $\mathcal{E}_B$ and the given bound on $N$ it holds that

$$\begin{aligned}
d^{\mathsf{b}}(u, ao) - \frac{M}{N} &\leq G_\delta(M/N, N) \\
\Leftrightarrow \quad 0 &\leq M + NG_\delta(M/N, N) - Nd^{\mathsf{b}}(u, ao) \\
&\leq M + \sqrt{2M \log(4/\delta)} + 14 \log(4/\delta)/3 - \log(4/\delta)\left(2b + 31/6\right) \\
&= M + \sqrt{2 \log(4/\delta)}\sqrt{M} - \log(4/\delta)\left(2b + 1/2\right).
\end{aligned}$$

Solving the quadratic inequality for positive $\sqrt{M}$ yields

$$\begin{aligned}
\sqrt{M} &\geq -\sqrt{\frac{\log(4/\delta)}{2}} + \sqrt{\frac{\log(4/\delta)}{2} + \log(4/\delta)\left(2b + 1/2\right)} \\
&= -\sqrt{\frac{\log(4/\delta)}{2}} + \sqrt{\log(4/\delta) + 2b \log(4/\delta)} \\
&\geq -\sqrt{\frac{\log(4/\delta)}{2}} + \frac{\sqrt{\log(4/\delta)} + \sqrt{2b \log(4/\delta)}}{\sqrt{2}} = \sqrt{b \log(4/\delta)},
\end{aligned}$$

where we have used the inequality $\sqrt{x+y} \geq (\sqrt{x} + \sqrt{y})/\sqrt{2}$. Hence the bound on $N$ in the lemma implies that $M = \sqrt{M}^2 \geq b \log(4/\delta)$. $\quad\square$

### B.1 Proof of Theorem 1

We first prove two lemmas very similar to Lemmas 16 and 17 of (Deb et al., 2025).

**Lemma 6.** *Let $\mathbf{R}$ be an RDP and let $\mathcal{X}$ be a language set that distinguishes $\mathbf{R}$ under the behavior policy $\pi^{\mathsf{b}}$. Given a candidate state $u, ao \in \mathcal{U} \times \mathcal{AO}$ and a reference state $u' \in \mathcal{U}$, let $\mathcal{Z}_1$ and $\mathcal{Z}_2$ be two multisets sampled from the true distributions $p_{uao}$ and $p_{u'}$ on suffixes in $\Gamma^+$, respectively. Under event $\mathcal{E}_\mathcal{X}$, if $\tau(u, ao) = u'$ then TESTDISTINCT($\mathcal{Z}_1, \mathcal{Z}_2, \delta$) returns false.*

*Proof.* Since $\tau(u, ao) = u'$, any pair of histories $h_1$ and $h_2$ associated with $u, ao$ and $u'$ satisfy $\bar{\tau}(h_1) = \bar{\tau}(h_2) = u'$. Since $\mathcal{X}$ distinguishes $\mathbf{R}$, this implies that $L_\mathcal{X}(p_{uao}, p_{u'}) = 0$ holds. Letting $\widehat{p}_{uao}$ and $\widehat{p}_{u'}$ be the empirical distributions on suffixes induced by the multisets $\mathcal{Z}_1$ and $\mathcal{Z}_2$, we can now use the event $\mathcal{E}_\mathcal{X}$, Lemma 2 and the triangle inequality to obtain

$$\begin{aligned}
L_\mathcal{X}(\widehat{p}_{uao}, \widehat{p}_{u'}) &\leq L_\mathcal{X}(\widehat{p}_{uao}, p_{uao}) + L_\mathcal{X}(p_{uao}, p_{u'}) + L_\mathcal{X}(p_{u'}, \widehat{p}_{u'}) \\
&\leq \sqrt{\frac{\log(2|\mathcal{X}|/\delta)}{2|\mathcal{Z}_1|}} + 0 + \sqrt{\frac{\log(2|\mathcal{X}|/\delta)}{2|\mathcal{Z}_2|}} \leq \sqrt{\frac{2 \log(2|\mathcal{X}|/\delta)}{\min(|\mathcal{Z}_1|, |\mathcal{Z}_2|)}}.
\end{aligned}$$

This is precisely the condition for which TESTDISTINCT returns false. $\quad\square$

**Lemma 7.** *Let $\mathbf{R}$ be an RDP and let $\mathcal{X}$ be a language set that distinguishes $\mathbf{R}$ under the behavior policy $\pi^{\mathsf{b}}$. Given a candidate state $u, ao \in \mathcal{U} \times \mathcal{AO}$ and a reference state $u' \in \mathcal{U}$, let $\mathcal{Z}_1$ and $\mathcal{Z}_2$ be two multisets sampled from the true distributions $p_{uao}$ and $p_{u'}$ on suffixes in $\Gamma^+$, respectively. Under event $\mathcal{E}_{\mathcal{X}}$, if $\tau(u, ao) \neq u'$ then $\textsc{TestDistinct}(\mathcal{Z}_1, \mathcal{Z}_2, \delta)$ answers true if $\mathcal{Z}_1$ and $\mathcal{Z}_2$ satisfy $\min(|\mathcal{Z}_1|, |\mathcal{Z}_2|) \geq 8 \log(2|\mathcal{X}|/\delta)/\mu_{\mathcal{X}}^2$.*

*Proof.* Since $\tau(u, ao) \neq u'$, any pair of histories $h_1$ and $h_2$ associated with $u, ao$ and $u'$ satisfy $\bar{\tau}(h_1) \neq \bar{\tau}(h_2)$. Since $\mathcal{X}$ distinguishes $\mathbf{R}$, this implies that $L_{\mathcal{X}}(p_{uao}, p_{u'}) \geq \mu_{\mathcal{X}}$ holds. Letting $\widehat{p}_{uao}$ and $\widehat{p}_{u'}$ be the empirical distributions on suffixes induced by the multisets $\mathcal{Z}_1$ and $\mathcal{Z}_2$, we can now use the event $\mathcal{E}_{\mathcal{X}}$, Lemma 2 and the triangle inequality to obtain

$$L_{\mathcal{X}}(\widehat{p}_{uao}, \widehat{p}_{u'}) \geq L_{\mathcal{X}}(p_{uao}, p_{u'}) - L_{\mathcal{X}}(\widehat{p}_{uao}, p_{uao}) - L_{\mathcal{X}}(p_{u'}, \widehat{p}_{u'})$$

$$\geq \mu_{\mathcal{X}} - \sqrt{\frac{\log(2|\mathcal{X}|/\delta)}{2|\mathcal{Z}_1|}} - \sqrt{\frac{\log(2|\mathcal{X}|/\delta)}{2|\mathcal{Z}_2|}}$$

$$\geq \mu_{\mathcal{X}} - \sqrt{\frac{2 \log(2|\mathcal{X}|/\delta)}{\min(|\mathcal{Z}_1|, |\mathcal{Z}_2|)}} \geq \mu_{\mathcal{X}} - \sqrt{\frac{\mu_{\mathcal{X}}^2}{4}} = \frac{\mu_{\mathcal{X}}}{2} \geq \sqrt{\frac{2 \log(2|\mathcal{X}|/\delta)}{\min(|\mathcal{Z}_1|, |\mathcal{Z}_2|)}},$$

where we have used the given condition on $\min(|\mathcal{Z}_1|, |\mathcal{Z}_2|)$ twice on the last line. This is precisely the condition for which $\textsc{TestDistinct}$ returns true. $\qquad\square$

The following lemma shows that the algorithm ADACT–L returns a minimal RDP if the multisets $\mathcal{Z}$ associated with candidate states satisfy $|\mathcal{Z}| \geq 16 \log(4/\delta) \log |\mathcal{X}|/\mu_{\mathcal{X}}^2 \equiv M_{\mathcal{X}}$.

**Lemma 8.** *Under event $\mathcal{E}_{\mathcal{X}}$, ADACT–L outputs a minimal automaton $\mathbf{A}_{\mathsf{r}}$ if the language set $\mathcal{X}$ distinguishes $\mathbf{A}_{\mathsf{p}} \ltimes \mathbf{A}_{\mathsf{r}}$ under the behavior policy $\pi^{\mathsf{b}}$ and the multiset $\mathcal{Z}(uao)$ associated with each candidate state $u, ao \in \mathcal{U} \times \mathcal{AO}$ satisfies $|\mathcal{Z}(uao)| \geq M_{\mathcal{X}}$.*

*Proof.* We prove the lemma using induction on RDP states $u = u_{\mathsf{p}} u_{\mathsf{r}} \in \mathcal{U}_{\mathsf{p}} \mathcal{U}_{\mathsf{r}}$. Since the algorithm uses a queue data structure, such state pairs are visited in breadth-first order. The base case is given by the initial state pair $u_{\mathsf{p}}^0 u_{\mathsf{r}}^0$ and the associated multiset $\mathcal{Z}(u_{\mathsf{p}}^0 u_{\mathsf{r}}^0) = \mathcal{D}$. This state pair is covered by the single initial state $u_{\mathsf{r}}^0$ that has to be part of any minimal automaton $\mathbf{A}_{\mathsf{r}}$.

The inductive case is given by a state pair $u_{\mathsf{p}} u_{\mathsf{r}}$ visited by the algorithm, and the associated multiset $\mathcal{Z}(u_{\mathsf{p}} u_{\mathsf{r}})$ induced by all shortest histories mapping to $u_{\mathsf{p}} u_{\mathsf{r}}$. By hypothesis of induction, all state pairs visited by the algorithm prior to (and including) $u_{\mathsf{p}} u_{\mathsf{r}}$ are induced by the known prior $\mathbf{A}_{\mathsf{p}}$ and a minimal automaton $\mathbf{A}_{\mathsf{r}}$. Consider an action-observation $ao \in \mathcal{AO}$ and let $\mathcal{Z}(ao)$ be the multiset of suffixes in $\mathcal{Z}(u_{\mathsf{p}} u_{\mathsf{r}})$ consistent with $ao$. Let $u_{\mathsf{p}}' = \tau_{\mathsf{p}}(u_{\mathsf{p}}, ao)$ be the resulting next state of the prior automaton, and let $u_{\mathsf{r}}' = \tau_{\mathsf{r}}(u_{\mathsf{r}}, u_{\mathsf{p}} ao)$ be the next state of a minimal automaton $\mathbf{A}_{\mathsf{r}}$. If $u_{\mathsf{p}}' u_{\mathsf{r}}'$ is visited before $u_{\mathsf{p}} u_{\mathsf{r}}$, then Lemma 6 implies that $\textsc{TestDistinct}(\mathcal{Z}(ao), \mathcal{Z}(u_{\mathsf{p}}' u_{\mathsf{r}}'), \delta)$ returns false. In this case the algorithm correctly defines $\tau_{\mathsf{r}}(u_{\mathsf{r}}, u_{\mathsf{p}} ao) = u_{\mathsf{r}}'$, and does not enqueue a new state pair. On the other hand, if $u_{\mathsf{p}}' u_{\mathsf{r}}'$ is not visited before $u_{\mathsf{p}} u_{\mathsf{r}}$, then if the multisets associated with all candidate states have cardinality at least $M_{\mathcal{X}}$, Lemma 7 implies that $\textsc{TestDistinct}(\mathcal{Z}(ao), \mathcal{Z}(\hat{u}_{\mathsf{p}} \hat{u}_{\mathsf{r}}), \delta)$ returns true for all state pairs $\hat{u}_{\mathsf{p}} \hat{u}_{\mathsf{r}}$ visited before $u_{\mathsf{p}} u_{\mathsf{r}}$. In this case the algorithm defines $\tau_{\mathsf{r}}(u_{\mathsf{r}}, u_{\mathsf{p}} ao) = u_{\mathsf{r}}'$ for the next available state $u_{\mathsf{r}}' \in \mathcal{U}_{\mathsf{r}}$ associated with $u_{\mathsf{p}}'$, and enqueues a new state pair $u_{\mathsf{p}}' u_{\mathsf{r}}'$. This proves that the output of the algorithm is the transition function $\tau_{\mathsf{r}}$ of a minimal automaton $\mathbf{A}_{\mathsf{r}}$. $\qquad\square$

To complete the proof of the theorem we need to select a minimum number of episodes to ensure that $|\mathcal{Z}(uao)| \geq M_{\mathcal{X}}$ for each $u, ao$. Choosing $b = 16 \log |\mathcal{X}|/\mu_{\mathcal{X}}^2$ in Lemma 5, we get the following bound:

$$N \geq \max_{u, ao} \left\{ \frac{\log(4/\delta)}{d^{\mathsf{b}}(u, ao)} \left( \frac{32 \log |\mathcal{X}|}{\mu_{\mathcal{X}}^2} + 31/6 \right) \right\}.$$

Since $\mathcal{X}$ distinguishes $\mathbf{A}_{\mathsf{p}} \ltimes \mathbf{A}_{\mathsf{r}}$ and event $\mathcal{E}_{\mathcal{X}}$ holds, Lemma 8 now directly applies. It is sufficient to choose $\delta_0 = \delta/2UU_{\mathsf{p}}AO$ to ensure that events $\mathcal{E}_{\mathcal{X}}$ and $\mathcal{E}_B$ hold for all candidate states. Using the lower bound $d^{\mathsf{b}}(u, ao) \geq d^*(u, ao)/C_{\mathbf{R}}^* \geq d_{\mathsf{m}}^*/C_{\mathbf{R}}^*$ yields

$$N \geq \frac{C_{\mathbf{R}}^* \log(8UU_{\mathsf{p}}AO/\delta_0)}{d_{\mathsf{m}}^*} \left( \frac{32 \log |\mathcal{X}|}{\mu_{\mathcal{X}}^2} + 31/6 \right) = \widetilde{\mathcal{O}} \left( \frac{C_{\mathbf{R}}^* \log(1/\delta) \log |\mathcal{X}|}{d_{\mathsf{m}}^* \cdot \mu_{\mathcal{X}}^2} \right).$$

which concludes the proof. We remark that Deb et al. (2025) present an improved analysis for an approximate version of their algorithm, but we leave a similar analysis for future work.

## C  APPROXIMATION ALGORITHM

In this appendix we prove a sample complexity bound for the approximation algorithm ADACT–L–A presented below. The algorithm is identical to ADACT–L, but if the multiset of a candidate state is smaller than a given threshold (line 10), the candidate state maps to an absorbing dummy state. The resulting RDP $\mathbf{A}_\mathsf{p} \ltimes \mathbf{A}_\mathsf{r}'$ approximates the minimal RDP $\mathbf{A}_\mathsf{p} \ltimes \mathbf{A}_\mathsf{r}$, and the threshold is selected such that an optimal policy for $\mathbf{A}_\mathsf{p} \ltimes \mathbf{A}_\mathsf{r}'$ is an $\varepsilon/2$-approximation of the optimal policy for $\mathbf{A}_\mathsf{p} \ltimes \mathbf{A}_\mathsf{r}$.

---

**Function** ADACT–L–A$(\mathcal{U}_\mathsf{r}, \mathcal{D}, \mathbf{A}_\mathsf{p}, \delta, \overline{U}, \overline{C})$

**Input:** Automaton states $\mathcal{U}_\mathsf{r} = \{u_\mathsf{r}^\perp, u_\mathsf{r}^0, u_\mathsf{r}^1, u_\mathsf{r}^2, \dots\}$, dataset $\mathcal{D}$ of traces in $\Gamma^{H+1}$,
         prior automaton $\mathbf{A}_\mathsf{p} = \langle \Sigma, \mathcal{U}_\mathsf{p}, \tau_\mathsf{p}, u_\mathsf{p}^0 \rangle$, failure probability $0 < \delta < 1$, upper bounds $\overline{U}$ and $\overline{C}$
**Output:** Transition function $\tau_\mathsf{r}' : \mathcal{U}_\mathsf{r} \times \mathcal{U}_\mathsf{p}\,\mathcal{A}\mathcal{O} \to \mathcal{U}_\mathsf{r}$

1 **foreach** $u_\mathsf{p} \in \mathcal{U}_\mathsf{p}$ **do** $i(u_\mathsf{p}) \leftarrow 0$
2 **foreach** $u_\mathsf{p}ao \in \mathcal{U}_\mathsf{p}\mathcal{A}\mathcal{O}$ **do** $\tau_\mathsf{r}'(u_\mathsf{r}^\perp, u_\mathsf{p}ao) \leftarrow u_\mathsf{r}^\perp$
3 $Q \leftarrow \{u_\mathsf{p}^0 u_\mathsf{r}^0\}$                  // queue data structure containing $u_\mathsf{p}^0 u_\mathsf{r}^0$
4 $i(u_\mathsf{p}^0) \leftarrow 1$, $t(u_\mathsf{p}^0 u_\mathsf{r}^0) \leftarrow 0$, $\mathcal{Z}(u_\mathsf{p}^0 u_\mathsf{r}^0) \leftarrow \mathcal{D}$
5 **while** $Q$ *is not empty* **do**
6    dequeue $u_\mathsf{p}u_\mathsf{r}$ from $Q$                      // next joint state
7    **for** $ao \in \mathcal{A}\mathcal{O}$ **do**
8      $u_\mathsf{p}' \leftarrow \tau_\mathsf{p}(u_\mathsf{p}, ao)$                // next prior state
9      $\mathcal{Z}(ao) \leftarrow \{e_{t+1:H} \mid ao/re_{t+1:H} \in \mathcal{Z}(u_\mathsf{p}u_\mathsf{r})\}$      // compute suffixes
10      **if** $|\mathcal{Z}(ao)|/|\mathcal{D}| < 3\varepsilon/(10\overline{U}AO\overline{C})$ **then**
11        $\tau_\mathsf{r}'(u_\mathsf{r}, u_\mathsf{p}ao) \leftarrow u_\mathsf{r}^\perp$            // map to dummy state
12      **else**
13        $j \leftarrow i(u_\mathsf{p}')$
14        **for** $k = 0, \dots, i(u_\mathsf{p}') - 1$ **do**
15          **if** *not* TESTDISTINCT$(\mathcal{Z}(u_\mathsf{p}'u_\mathsf{r}^k), \mathcal{Z}(ao), \delta)$ **then** $j \leftarrow k$
16        **end**
17        $\tau_\mathsf{r}'(u_\mathsf{r}, u_\mathsf{p}ao) \leftarrow u_\mathsf{r}^j$           // define transition function
18        **if** $j = i(u_\mathsf{p}')$ **then**
19          enqueue $u_\mathsf{p}'u_\mathsf{r}^j$ in $Q$
20          $i(u_\mathsf{p}') \leftarrow j + 1$, $t(u_\mathsf{p}'u_\mathsf{r}^j) \leftarrow t(u_\mathsf{p}u_\mathsf{r}) + 1$, $\mathcal{Z}(u_\mathsf{p}'u_\mathsf{r}^j) \leftarrow \mathcal{Z}(ao)$
21        **else if** $t(u_\mathsf{p}'u_\mathsf{r}^j) = t(u_\mathsf{p}u_\mathsf{r}) + 1$ **then** $\mathcal{Z}(u_\mathsf{p}'u_\mathsf{r}^j) \leftarrow \mathcal{Z}(u_\mathsf{p}'u_\mathsf{r}^j) \cup \mathcal{Z}(ao)$
22      **end**
23    **end**
24 **end**
25 **return** $\tau_\mathsf{r}'$
26 **Function** TESTDISTINCT$(\mathcal{Z}_1, \mathcal{Z}_2, \delta)$
27    **return** $L_\mathcal{X}(\mathcal{Z}_1, \mathcal{Z}_2) \geq \sqrt{\log(2|\mathcal{X}|/\delta)/\min(|\mathcal{Z}_1|, |\mathcal{Z}_2|)}$       // statistical test

---

Concretely, the subroutine TESTDISTINCT is only called for a candidate state $uao$ on line 15 when $\widehat{p}(uao)$ satisfies

$$\widehat{p}(uao) = \frac{|\mathcal{Z}(ao)|}{|\mathcal{D}|} \geq \frac{3\varepsilon}{10\overline{U}AO\overline{C}} \equiv \psi,$$

where $\varepsilon, \overline{U}$ and $\overline{C}$ are inputs to the algorithm and $\psi$ is the threshold. We prove the following theorem:

**Theorem 9.** *With probability at least $1 - 2AO\overline{U}U_\mathsf{p}\delta$, ADACT-L-A$(\mathcal{U}_\mathsf{r}, \mathcal{D}, \mathbf{A}_\mathsf{p}, \delta, \overline{U}, \overline{C})$ returns an automaton $\mathbf{A}_\mathsf{r}'$ such that $\mathbf{A}_\mathsf{p} \ltimes \mathbf{A}_\mathsf{r}'$ is an $\frac{\varepsilon}{2}$-approximation of the minimal RDP $\mathbf{A}_\mathsf{p} \ltimes \mathbf{A}_\mathsf{r}$ when using a language set $\mathcal{X}$ that distinguishes $\mathbf{A}_\mathsf{p} \ltimes \mathbf{A}_\mathsf{r}$ under the behavior policy $\pi^\mathsf{b}$ with associated distinguishability $\mu_\mathcal{X}$ and the size of the dataset $\mathcal{D}$ is at least*

$$|\mathcal{D}| \geq \widetilde{\mathcal{O}} \left( \frac{\overline{U}AO\overline{C} \log(1/\delta) \log|\mathcal{X}|}{\varepsilon \mu_\mathcal{X}^2} \right).$$

We first prove that the resulting RDP $\mathbf{R}' = \mathbf{A}_\mathsf{p} \ltimes \mathbf{A}_\mathsf{r}'$ is $\frac{\varepsilon}{2}$-approximate.

**Lemma 10.** *Under events $\mathcal{E}_\mathcal{X}$ and $\mathcal{E}_B$, if $\overline{U}$ and $\overline{C}$ are upper bounds on the number of RDP states $|\mathcal{U}_\mathsf{r}|$ and concentrability $C^*_{\mathbf{R}'}$ of the resulting RDP $\mathbf{R}' = \mathbf{A}_\mathsf{p} \ltimes \mathbf{A}_\mathsf{r}'$, then ADACT-H-A returns an returns an automaton $\mathbf{A}_\mathsf{r}'$ such that $\mathbf{R}'$ is an $\frac{\varepsilon}{2}$-approximation of the minimal RDP $\mathbf{R} = \mathbf{A}_\mathsf{p} \ltimes \mathbf{A}_\mathsf{r}$.*

*Proof.* Consider a candidate state $uao$ with $M = |\mathcal{Z}(ao)|$. If $\widehat{p}(uao) \geq \psi$ we impose the condition $M \geq M_\mathcal{X}$ as before. For each such candidate state, ADACT–L–A calls TESTDISTINCT and correctly promotes $uao$ to an automaton state or merges it with an existing automaton state.

On the other hand, if $\widehat{p}(uao) < \psi$ and $N \geq 16 \log(4/\delta)/\psi$, event $\mathcal{E}_B$ and Lemma 3 yield

$$d_t^\mathsf{b}(u, ao) - \widehat{p}(uao) \leq G_\delta(\widehat{p}(qao), N)$$

$$\Leftrightarrow \quad d_t^\mathsf{b}(u, ao) < \widehat{p}(uao) + G_\delta(\widehat{p}(uao), N) < \psi + G_\delta(\psi, N) \leq \frac{5\psi}{3} = \frac{\varepsilon}{2\overline{U} AO\overline{C}}.$$

In this case, ADACT–L–A does not call TESTDISTINCT and hence the resulting RDP state may be incorrect. We can bound the contribution of $uao$ to the value under the optimal policy $\pi^*$ as

$$d_t^*(u, ao) \sum_{a' \in \mathcal{A}} \pi^*(\tau(u, ao), a') \sum_{r \in \mathcal{R}} \theta_\mathsf{r}(\tau(u, ao), a', r) \cdot r$$

$$\leq d_t^*(u, ao) \sum_{a' \in \mathcal{A}} \pi^*(\tau(u, ao), a') \sum_{r \in \mathcal{R}} \theta_\mathsf{r}(\tau(u, ao), a', r) = d_t^*(u, ao) \leq C^*_{\mathbf{R}'} d_t^\mathsf{b} \leq \frac{\varepsilon}{2\overline{U} AO},$$

where we have used the fact that the reward is bounded by $1$. Summing up the contribution of all such incorrect candidate states to the expected optimal value of histories in $\mathcal{H}$ yields

$$\sum_{t \in [\![0,H]\!]} \sum_{u_t ao} d_t^*(u_t, ao) \sum_{a' \in \mathcal{A}} \pi^*(\tau(u, ao), a') \sum_{r \in \mathcal{R}} \theta_\mathsf{r}(\tau(u, ao), a', r) \cdot r \leq \sum_{t \in [\![0,H]\!]} \sum_{u_t ao} \frac{\varepsilon}{2\overline{U} AO} \leq \frac{\varepsilon}{2}.$$

This proves that the resulting RDP $\mathbf{R}'$ is $\frac{\varepsilon}{2}$-approximate. $\qquad\square$

To prove Theorem 9, for each candidate state $uao$ such that $\widehat{p}(uao) < \psi$, a number of episodes which satisfies $N \geq 16 \log(4/\delta)/\psi$ is sufficient to ensure that $\mathbf{R}'$ is $\frac{\varepsilon}{2}$-approximate. If $\widehat{p}(uao) \geq \psi$, we instead require $M \geq M_\mathcal{X}$ as before. Since $M_\mathcal{X} = 16 \log(4/\delta) \log |\mathcal{X}|/\mu_\mathcal{X}^2$, event $\mathcal{E}_B$ together with Corollary 4 yield

$$\widehat{p}(uao) - d_t^\mathsf{b}(u, ao) \leq \frac{2\widehat{p}(uao)}{3} \quad \Leftrightarrow \quad d_t^\mathsf{b}(u, ao) \geq \frac{\widehat{p}(uao)}{3} \geq \frac{\psi}{3} = \frac{\varepsilon}{10\overline{U} AO\overline{C}}.$$

Choosing $b = 16 \log |\mathcal{X}|/\mu_\mathcal{X}^2$ in Lemma 5 and enforcing $N \geq 16 \log(4/\delta)/\psi$ yields

$$N \geq \max_{uao} \left\{ \frac{\log(4/\delta)}{d_t^\mathsf{b}(u, ao)} \left( \frac{32 \log |\mathcal{X}|}{\mu_\mathcal{X}^2} + 31/6 \right) \right\} + \frac{16 \log(4/\delta)}{\psi}.$$

We can now use the definition of $\psi$ and the lower bound on $d_t^\mathsf{b}(u, ao)$ in the case $\widehat{p}(uao) \geq \psi$ to achieve the following bound:

$$N \geq \frac{10\overline{U} AO\overline{C} \log(8\overline{U} U_\mathsf{p} AO/\delta_0)}{\varepsilon} \left( \frac{32 \log |\mathcal{X}|}{\mu_\mathcal{X}^2} + 31/6 \right) + \frac{160\overline{U} AO\overline{C} \log(8\overline{U} U_\mathsf{p} AO/\delta_0)}{3\varepsilon}$$

$$= \widetilde{\mathcal{O}} \left( \frac{\overline{U} AO\overline{C} \log(1/\delta) \log |\mathcal{X}|}{\varepsilon \mu_\mathcal{X}^2} \right).$$

## D  EXAMPLES

We provide several examples that help to understand important aspects of RDPs, as well as of our novel notions.

### D.1  EXAMPLE RDPS WITH A FOCUS ON DISTINGUISHABILITY

**Example 2.** Consider an RDP defined by $\mathbf{R} = \langle \mathcal{O}, \mathcal{A}, \mathcal{R}, \bar{T}, \bar{R}, H, \nu \rangle$ and $\mathbf{A} = \langle \mathcal{U}, \Sigma, \Omega, \tau, \theta, u_0 \rangle$ with components given by

$$\mathcal{O} = \{o_1, o_2\}, \quad \mathcal{A} = \{a_1, a_2\}, \quad \mathcal{R} = \{0, 1\}, \quad \mathcal{U} = \{u_0, u_1, u_2, u_3\}.$$

The (semi-)automaton $\mathbf{A}$ is illustrated in the following figure:

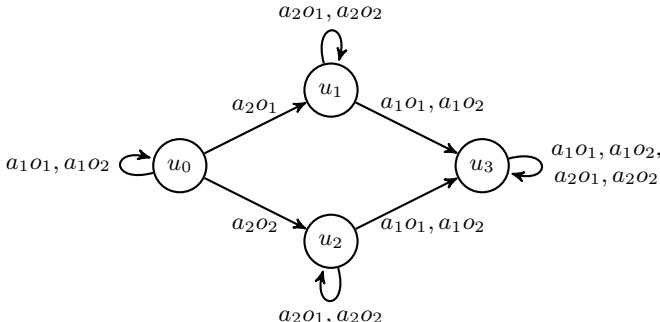

The output function $\theta$ is defined as follows:

- $\theta_{\mathsf{o}}(o \mid u, a) = 0.5$ for each $o \in \mathcal{O}$, $u \in \{u_0, u_3\}$ and $a \in \mathcal{A}$.

- $\theta_{\mathsf{o}}(o \mid u, a_2) = 0.5$ for each $o \in \mathcal{O}$ and $u \in \{u_1, u_2\}$.

- $\theta_{\mathsf{o}}(o_1 \mid u_1, a_1) = \theta_{\mathsf{o}}(o_2 \mid u_2, a_1) = 0.75$.

- $\theta_{\mathsf{o}}(o_2 \mid u_1, a_1) = \theta_{\mathsf{o}}(o_1 \mid u_2, a_1) = 0.25$.

- $\theta_{\mathsf{r}}(0 \mid u, a) = 1$ for each $u \in \{u_0, u_3\}$ and $a \in \mathcal{A}$.

- $\theta_{\mathsf{r}}(1 \mid u, a_1) = 1$ for each $u \in \{u_1, u_2\}$.

- $\theta_{\mathsf{r}}(0 \mid u, a_2) = 1$ for each $u \in \{u_1, u_2\}$.

Let $\pi$ be the regular policy defined as $\pi(a|u) = 0.5$ for each $a \in \mathcal{A}$ and each $u \in \mathcal{U}$. Let $X$ be the language defined by the regular expression $.*(.o_1 1).*$. Hence a string in $\Gamma^+ = (\mathcal{A}\mathcal{O}/\mathcal{R})^+$ belongs to $X$ if and only if the observation-reward pair $o_1 1$ appears in the string. Let $\mathcal{X} = \{X\}$ be the language set containing only $X$.

We claim that $\mathcal{X}$ distinguishes the RDP $\mathbf{R}$ under the regular policy $\pi$. For any history $h$ mapping to state $u_3$, the probability of the language $X$ is $p_h^\pi(X) = 0$ since the reward 1 can never appear. For any history $h$ mapping to state $u_1$, eventually the policy $\pi$ will select action $a_1$ and the probability of $o_1 1$ is $\theta_{\mathsf{o}}(o_1 \mid u_1, a_1)\theta_{\mathsf{r}}(1 \mid u_1, a_1) = 0.75 \cdot 1 = 0.75$, implying $p_h^\pi(X) = 0.75$. For any history $h$ mapping to state $u_2$, eventually the policy $\pi$ will select action $a_1$ and the probability of $o_1 1$ is $\theta_{\mathsf{o}}(o_1 \mid u_2, a_1)\theta_{\mathsf{r}}(1 \mid u_2, a_1) = 0.25 \cdot 1 = 0.25$, implying $p_h^\pi(X) = 0.25$. For any history $h$ mapping to state $u_0$, eventually the policy $\pi$ will select action $a_2$. This always causes a reward of 0 and transitions to $u_1$ or $u_2$ with equal probability. Hence the probability of $o_1 1$ is $0.5 \cdot 0.75 + 0.5 \cdot 0.25 = 0.5$, implying $p_h^\pi(X) = 0.5$.

As a consequence, given two histories $h, h' \in \mathcal{H}$, if $h \sim h'$ the language metric is given by $L_{\mathcal{X}}(p_h^\pi, p_{h'}^\pi) = |p_h^\pi(X) - p_{h'}^\pi(X)| = 0$, while if $h \not\sim h'$ we have $L_{\mathcal{X}}(p_h^\pi, p_{h'}^\pi) = |p_h^\pi(X) - p_{h'}^\pi(X)| \geq 0.25$. Hence $\mathcal{X}$ distinguishes $\mathbf{R}$ for $\pi$ and has distinguishability $\mu_{\mathcal{X}} = 0.25$. ∎

**Example 3.** Another example RDP is the following one.

$$\mathcal{O} = \{o_1, o_2\}, \quad \mathcal{A} = \{a_1, a_2\}, \quad \mathcal{R} = \{0, 1\}, \quad \mathcal{U} = \{u_0, u_1, u_2, u_3, u_4, u_5\}.$$

The (semi-)automaton $\mathbf{A}$ is illustrated in the following figure:

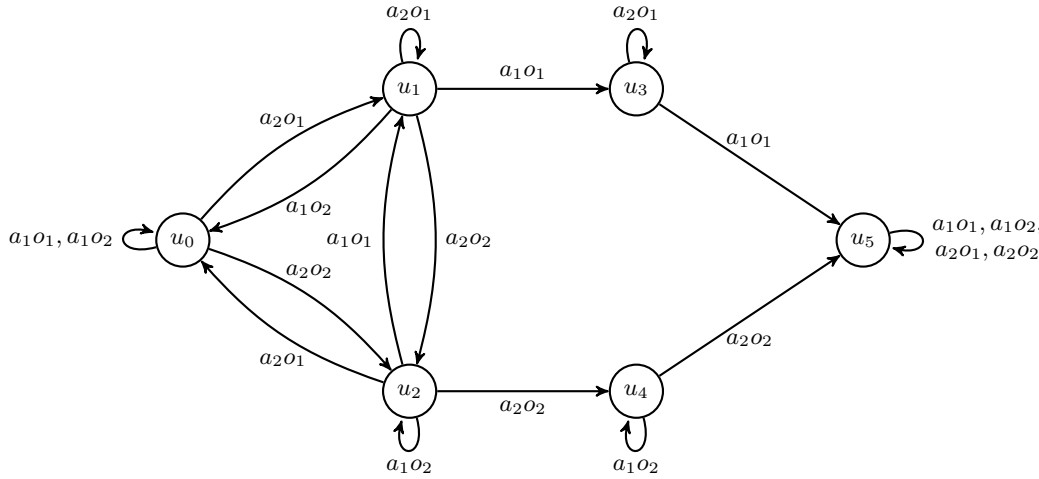

The output function $\theta$ is defined as follows:

- $\theta_{\mathsf{o}}(o \mid u, a) = 0.5$ for each $o \in \mathcal{O}$, $u \in \{u_0, u_5\}$ and $a \in \mathcal{A}$.

- $\theta_{\mathsf{o}}(o_1 \mid u_1, a) = \theta_{\mathsf{o}}(o_2 \mid u_2, a) = 0.75$ for each $a \in \mathcal{A}$.

- $\theta_{\mathsf{o}}(o_2 \mid u_1, a) = \theta_{\mathsf{o}}(o_1 \mid u_2, a) = 0.25$ for each $a \in \mathcal{A}$.

- $\theta_{\mathsf{o}}(o_1 \mid u_3, a) = \theta_{\mathsf{o}}(o_2 \mid u_4, a) = 1$ for each $a \in \mathcal{A}$.

- $\theta_{\mathsf{r}}(0 \mid u, a) = 1$ for each $u \in \{u_0, u_1, u_2, u_5\}$ and $a \in \mathcal{A}$.

- $\theta_{\mathsf{r}}(0 \mid u_3, a_2) = \theta_{\mathsf{r}}(0 \mid u_4, a_1) = 1$.

- $\theta_{\mathsf{r}}(1 \mid u_3, a_1) = \theta_{\mathsf{r}}(1 \mid u_4, a_2) = 1$.

Consider the regular policy $\pi$ defined as $\pi(a|u) = 0.5$ for each $u \in \mathcal{U}$ and $a \in \mathcal{A}$. Some facts about the RDP:

- From state $u_5$ we can never observe reward 1.

- From state $u_3$ we eventually observe $o_1 1$.

- From state $u_4$ we eventually observe $o_2 1$.

- From state $u_1$ we eventually reach $u_3$ with probability 0.75 and $u_4$ with probability 0.25.

- From state $u_2$ we eventually reach $u_3$ with probability 0.25 and $u_4$ with probability 0.75.

- From state $u_0$ we eventually reach $u_3$ with probability 0.5 and $u_4$ with probability 0.5.

To prove the last three facts, let $p_0$, $p_1$, $p_2$ be the probability of reaching $u_3$ from $u_0$, $u_1$, $u_2$ respectively. These probabilities satisfy the following system of linear equations:

$$p_0 = 0.5p_1 + 0.5p_2,$$
$$p_1 = 0.2p_0 + 0.2p_2 + 0.6,$$
$$p_2 = 0.2p_0 + 0.2p_1.$$

The solution is given by $p_0 = 0.5$, $p_1 = 0.75$, $p_2 = 0.25$.

Consider the language set $\mathcal{X} = \{X_1, X_2\}$, where $X_1$ is the language defined by the regular expression `.*(.o_1 1).*` and $X_2$ is the language defined by the regular expression `.*(.o_2 1).*`. For each state $u \in \mathcal{U}$, the probabilities of the two languages for histories $h$ that map to $u$, i.e. $\bar{\tau}(h) = u$, are given by

$$
\begin{array}{lll}
u_0: & p_h^\pi(X_1) = 0.5, & p_h^\pi(X_2) = 0.5, \\
u_1: & p_h^\pi(X_1) = 0.75, & p_h^\pi(X_2) = 0.25, \\
u_2: & p_h^\pi(X_1) = 0.25, & p_h^\pi(X_2) = 0.75, \\
u_3: & p_h^\pi(X_1) = 1, & p_h^\pi(X_2) = 0, \\
u_4: & p_h^\pi(X_1) = 0, & p_h^\pi(X_2) = 1, \\
u_5: & p_h^\pi(X_1) = 0, & p_h^\pi(X_2) = 0.
\end{array}
$$

It is easy to verify that for the given language set $\mathcal{X}$ and two histories $h, h' \in \mathcal{H}$, $L_{\mathcal{X}}(p_h^\pi, p_{h'}^\pi) = 0$ if $h \sim h'$ and $L_{\mathcal{X}}(p_h^\pi, p_{h'}^\pi) \geq 0.25$ if $h \nsim h'$. Hence $\mathcal{X}$ distinguishes $\mathbf{R}$ and the distinguishability is $\mu_{\mathcal{X}} = 0.25$.

We can represent the RDP more compactly using a cascade $\mathbf{A}_o \ltimes \mathbf{A}_r$, where $\mathbf{A}_o$ is a Markov prior and $\mathbf{A}_r$ is the following automaton:

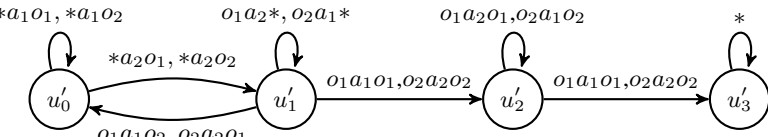

Concretely, the state $o_1 u_1'$ in the cascade corresponds to the state $u_1$ in the original RDP, while $o_2 u_1'$ corresponds to $u_2$. Likewise, $o_1 u_2'$ in the cascade corresponds to the state $u_3$ in the original RDP, while $o_2 u_2'$ corresponds to $u_4$. Both $o_1 u_0'$ and $o_2 u_0'$ map to $u_0$, and both $o_1 u_3'$ and $o_2 u_3'$ map to $u_5$. Note that the automaton $\mathbf{A}_r$ is more compact than the original RDP. ∎

**Example 4.** A third example to illustrate the difficulty of suffixes with different lengths. Here I have omitted actions and observations and focus only on probability distributions over suffixes (under the given behavior policy). For simplicity, assume that all transitions are deterministic except for $u_2 \to u_3$, which has probability $p$ (else the agent remains in $u_2$).

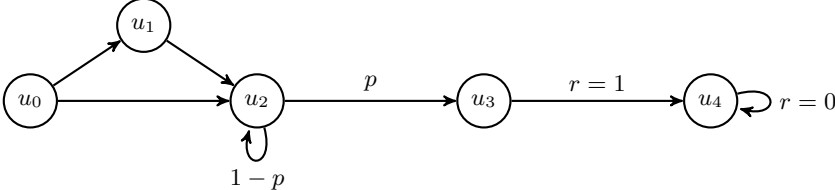

We can reach $u_2$ in two different ways: directly from $u_0$ (history $h$), or via $u_1$ (history $h'$). Let us assume that the only language in $\mathcal{X}$ checks if reward 1 is present in a suffix. The current algorithm will estimate $L_{\mathcal{X}}(p_h^\pi, p_{h'}^\pi)$ using two multisets of suffixes: one whose suffixes have length $H - 1$, and one whose suffixes have length $H - 2$.

The probability of *not* reaching $u_3$ in $k$ steps is $(1-p)^k$, since the agent will attempt to reach $u_3$ every timestep and fails with probability $1 - p$. Hence the probability of observing reward 1 in suffixes of length $H - 1$ is $1 - (1-p)^{H-2}$, and the probability of observing reward 1 in suffixes of length $H - 2$ is $1 - (1-p)^{H-3}$. To observe reward 1 in $k$ steps we have to reach $u_3$ in $k - 1$ steps to have time for the last transition from $u_3$ to $u_4$. For example, if $p = 0.1$ and $H = 10$ we have

$$
1 - (1-p)^{H-2} = 1 - 0.9^8 = 0.57,
$$
$$
1 - (1-p)^{H-3} = 1 - 0.9^7 = 0.52.
$$

∎

## D.2 EXAMPLES FOR SECTION 2 (PRELIMINARIES)

**Example 5.** Specific policies may induce the same distribution for histories that are not equivalent, as noted in Observation 1. This phenomenon can be observed in the following example, which focuses on the probability of observations, omitting rewards since they follow the same argument,

$$\mathcal{A} = \{a_1, a_2\}, \quad \mathcal{O} = \{o_1, o_2\}, \quad \mathcal{U} = \{u_0, u_1, u_2\},$$

$$\tau(u_0, ao_1) = u_1 \ \forall a \in \mathcal{A}, \quad \tau(u_0, ao_2) = u_2 \ \forall a \in \mathcal{A}, \quad \tau(u_i, ao) = u_i \ \forall ao \in \mathcal{AO}, \forall i \in \{1, 2\},$$

$$\theta_{\mathsf{o}}(o_1 \,|\, u_1, a_1) = 0.1, \quad \theta_{\mathsf{o}}(o_1 \,|\, u_1, a_2) = 0.9, \quad \theta_{\mathsf{o}}(o_2 \,|\, u_1, a_1) = 0.9, \quad \theta_{\mathsf{o}}(o_2 \,|\, u_1, a_2) = 0.1,$$

$$\theta_{\mathsf{o}}(o_1 \,|\, u_2, a_1) = 0.9, \quad \theta_{\mathsf{o}}(o_1 \,|\, u_2, a_2) = 0.1, \quad \theta_{\mathsf{o}}(o_2 \,|\, u_2, a_1) = 0.1, \quad \theta_{\mathsf{o}}(o_2 \,|\, u_2, a_2) = 0.9.$$

In this example, a regular policy causing the collapse of distributions over observations determined by the two different states $u_1, u_2$ is the following one, defined as a function of RDP states,

$$\pi(a_1 \,|\, u_1) = 0.9, \quad \pi(a_2 \,|\, u_1) = 0.1, \quad \pi(a_1 \,|\, u_2) = 0.1, \quad \pi(a_2 \,|\, u_2) = 0.9.$$

For instance, we have that the probability of $o_1$ coincides in the two states $u_1$ and $u_2$,

$$\mathbb{P}(o_1 \,|\, u_1, \pi) = \theta_{\mathsf{o}}(o_1 \,|\, u_1, a_1) \cdot \pi(a_1 \,|\, u_1) + \theta_{\mathsf{o}}(o_1 \,|\, u_1, a_2) \cdot \pi(a_2 \,|\, u_1) = 0.18,$$

$$\mathbb{P}(o_1 \,|\, u_1, \pi) = \theta_{\mathsf{o}}(o_1 \,|\, u_1, a_1) \cdot \pi(a_1 \,|\, u_1) + \theta_{\mathsf{o}}(o_1 \,|\, u_1, a_2) \cdot \pi(a_2 \,|\, u_1) = 0.18.$$

Similarly for $o_2$, we have $\mathbb{P}(o_2 \,|\, u_1, \pi) = \mathbb{P}(o_2 \,|\, u_2, \pi) = 0.82$. In general $p_{h_1}^\pi = p_{h_2}^\pi$ for histories $h_1, h_2$ mapping to $u_1, u_2$ respectively, even though $h_1 \not\sim h_2$ since $u_1 \neq u_2$. ∎

## D.3 EXTENDED VERSION OF EXAMPLE 1 (PARTIAL INDEPENDENCE FROM PRIORS)

The T-maze with corridor length $N$ and horizon $H$ has observations, actions, and rewards given by

$$\mathcal{O} = \{InCorridor, InJunction, GoalNorth, GoalSouth\},$$

$$\mathcal{A} = \{North, South, East, West\},$$

$$\mathcal{R} = \{0, 1\},$$

$$\mathcal{U} = \big(\{corridor\} \times [\![0, N]\!] \cup \{junction\} \times [\![-1, +1]\!]\big) \times \{GoalNorth, GoalSouth\},$$

and it is represented by the cascade $\mathbf{T}_H \times \mathbf{A}$ where $\mathbf{T}_H$ is the timestep prior and the semiautomaton $\mathbf{A} = \langle \mathcal{U}, \mathcal{AO}, \tau, u_0 \rangle$ is defined as follows.

States,

$$\mathcal{U} = \{u_0\} \cup \big((\{corridor\} \times [\![0, N]\!] \cup \{junction\} \times [\![-1, +1]\!]) \times \{GoalNorth, GoalSouth\}\big).$$

The transition function is defined as follows, where all variables range over their entire respective domains,

$$\tau(u_0, a\, goal) = \begin{cases} \langle corridor, 1, goal \rangle & \text{if } a = West \\ \langle corridor, 0, goal \rangle & \text{otherwise} \end{cases}$$

$$\tau(corridor, x, goal, ao) = \begin{cases} \langle corridor, x, goal \rangle & \text{if } a = North \text{ or } a = South \\ \langle corridor, \max(0, x-1), goal \rangle & \text{if } a = East \\ \langle corridor, x+1, goal \rangle & \text{if } a = West \text{ and } x < N \\ \langle junction, 0, goal \rangle & \text{if } a = West \text{ and } x = N \end{cases}$$

$$\tau(junction, y, goal, ao) = \begin{cases} \langle junction, \min(1, y+1), goal \rangle & \text{if } a = North \\ \langle junction, \max(-1, y-1), goal \rangle & \text{if } a = South \\ \langle junction, y, goal \rangle & \text{if } a = West \\ \langle junction, y, goal \rangle & \text{if } a = East \text{ and } y \neq 0 \\ \langle corridor, N, goal \rangle & \text{if } a = East \text{ and } y = 0 \end{cases}$$

Let $\perp$ represents the observation symbol marking the end of an episode. When the symbol is produced, the generated episode trace is to be considered complete.

The (deterministic) observation output function over the cascade state space $\theta_{\mathsf{o}} : [\![0, H]\!] \times \mathcal{U} \times \mathcal{A} \rightarrow \mathcal{O}$ is defined as follows, where $t$ ranges over $[\![0, H-1]\!]$.

$$\theta_{\mathsf{o}}(t, corridor, x, goal, a) = \begin{cases} corridor & \text{if } a = North \text{ or } a = South \\ corridor & \text{if } a = East \\ corridor & \text{if } a = West \text{ and } x < N \\ junction & \text{if } a = West \text{ and } x = N \end{cases}$$

$$\theta_{\mathsf{o}}(t, junction, y, goal, a) = \begin{cases} junction & \text{if } a = North \\ junction & \text{if } a = South \\ junction & \text{if } a = West \\ junction & \text{if } a = East \text{ and } y \neq 0 \\ corridor & \text{if } a = East \text{ and } y = 0 \end{cases}$$

$$\theta_{\mathsf{o}}(H, u_1, u_2, u_3, a, o) = \perp$$

The (deterministic) reward output function over the cascade state space $\theta_{\mathsf{r}} : [\![0, H]\!] \times \mathcal{U} \times \mathcal{A} \rightarrow \mathcal{R}$ is defined as follows, where all variables range over their entire respective domains (including $t$),

$$\theta_{\mathsf{r}}(t, corridor, x, goal, a) = 0$$

$$\theta_{\mathsf{r}}(t, junction, y, goal, a) = \begin{cases} 1 & \text{if } y = 0 \text{ and } a = North \text{ and } goal = GoalNorth \\ 1 & \text{if } y = 0 \text{ and } a = South \text{ and } goal = GoalSouth \\ 0 & \text{otherwise} \end{cases}$$

**Showing partial independence from the timestep prior (semi-stationarity)** The automaton above already satisfies the cascade condition (I) since it is given by $\mathbf{T}_H \times \mathbf{A}$. We show its output functions satisfy conditions (II) and (III). The observation output function (seen as returning distributions) can be factored into the following two functions,

$$\theta_{\mathsf{o}}^{\mathsf{r}}(o \,|\, corridor, x, goal, a) = \begin{cases} 1 & \text{if } o = \theta_{\mathsf{o}}(corridor, x, goal, a) \\ 0 & \text{otherwise} \end{cases}$$

$$\theta_{\mathsf{o}}^{\mathsf{t}}(o \,|\, t, a) = \begin{cases} 1 & \text{if } o \neq \perp \text{ and} \\ 1 & \text{if } o = \perp \text{ and } t = H \\ 0 & \text{otherwise} \end{cases}$$

*Remark* 1. The above function $\theta_{\mathsf{o}}^{\mathsf{r}}$ does not specify episode termination, and hence, at learning time, the distributions it induces must be assessed by a language metric that ignores string length—as we do when relevant in our experiments.

The reward output function (seen as returning distributions) can be factored into the following functions, where all variables range over their entire respective domains,

$$\theta_{\mathsf{r}}^{\mathsf{r}}(r \,|\, corridor, x, goal, a) = 0$$

$$\theta_{\mathsf{r}}^{\mathsf{r}}(r \,|\, junction, y, goal, a) = \begin{cases} 1 & \text{if } r = \theta_{\mathsf{r}}(junction, y, goal, a) \\ 0 & \text{otherwise} \end{cases}$$

and

$$\theta_{\mathsf{r}}^{\mathsf{t}}(r \,|\, t, a) = 1.$$

The above shows that the T-maze is partially independent from the timestep prior, i.e., it is semi-stationary.

**Preserving rewards only**   In the T-maze automaton, we can also factor out a spatial prior as $(\mathbf{T}_H \times \mathbf{S}) \ltimes \mathbf{A}$ where $\mathbf{S}$ describes the space of the maze, using states

$$\mathcal{U}_\mathsf{s} = \{corridor\} \times [\![0, N]\!] \cup \{junction\} \times [\![-1, +1]\!].$$

Note that we could also use the spatial prior $\mathbf{G}_{3 \times (N+1)}$ (introduced earlier) as a correct over-approximation. However, introducing the independence $(\mathbf{T}_H \times \mathbf{S}) \times \mathbf{A}$ allows only for representing an approximation of the original automaton. Specifically, we can still represent the reward function exactly, clear from the fact that the function $\theta_\mathsf{r}$ above is independent of its first three arguments. However, we can no longer represent precisely distributions on observations, since the function $\theta_\mathsf{o}$ depends on its second and third arguments. The advantage is that the domain-specific automaton $\mathbf{A}$ is very compact. It only needs to remember the goal position communicated at the beginning of an episode, and it can do so by using the two states $\{GoalNorth, GoalSouth\}$.

# E    RDP LEARNING PIPELINE

Figure 1 portrays the learning pipeline for offline RL in RDPs.

The input dataset is split into two equal parts (Part 1 and Part 2). Part 1 will serve as input to ADACT–L. Part 2, together with the output of ADACT–L, will form the input to Markov Transformation, which output traces that adhere to the Markov property, as if they come from the equivalent MDP associated to the RDP **R**. These Markov transformed traces will be passed to any off-the-shelf algorithm for RL in episodic MDPs.

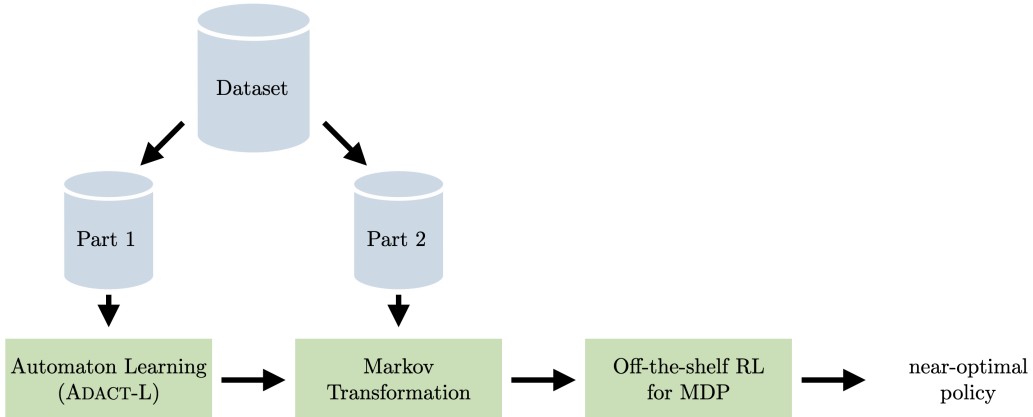

Figure 1: RDP Learning Pipeline

